# Hox genes regulate asexual reproductive behavior and tissue segmentation in adult animals

Christopher P. Arnold[1], Analí Migueles Lozano[2], Frederick G. Mann Jr [1], Stephanie H. Nowotarski[1], Julianna O. Haug [1], Jeffrey J. Lange[1], Chris W. Seidel[1] & Alejandro Sánchez Alvarado [1,3✉]

Hox genes are highly conserved transcription factors renowned for their roles in the segmental patterning of the embryonic anterior-posterior (A/P) axis. We report functions for Hox genes in A/P tissue segmentation and transverse fission behavior underlying asexual reproduction in adult planarian flatworms, *Schmidtea mediterranea*. Silencing of each of the *Hox* family members identifies 5 Hox genes required for asexual reproduction. Among these, silencing of *hox3* genes results in supernumerary fission segments, while silencing of *post2b* eliminates segmentation altogether. The opposing roles of *hox3* and *post2b* in segmentation are paralleled in their respective regulation of fission behavior. Silencing of *hox3* increases the frequency of fission behavior initiation while silencing of *post2b* eliminates fission behavior entirely. Furthermore, we identify a network of downstream effector genes mediating Hox gene functions, providing insight into their respective mechanisms of action. In particular, we resolve roles for *post2b* and effector genes in the functions of the marginal adhesive organ in fission behavior regulation. Collectively, our study establishes adult stage roles for Hox genes in the regulation of tissue segmentation and behavior associated with asexual reproduction.

[1] Stowers Institute, Kansas City, MO, USA. [2] University of Chicago, Chicago, IL, USA. [3] HHMI, Chevy Chase, MD, USA. ✉email: asa@stowers.org

We recently uncovered and characterized the size-dependent behaviors and adult tissue structures underlying asexual reproduction in the planarian, *Schmidtea mediterranea*[1] (Fig. 1a). These highly regenerative flatworms exist in both sexual and asexual biotypes. The latter reproduces solely through transverse fission – an asexual reproductive behavior in which torn posterior tissue fragments regenerate and give rise to clonal progeny[2]. Fission behavior is size-dependent, and its establishment and regulation coincide with growth-dependent patterning of the planarian central nervous system (CNS)[1]. Our study also revealed a cryptic form of segmentation that allocates tissue for fission progeny in coordination with the dynamic growth and de-growth of the adult planarian body plan[1,3]. Presently, the molecular mechanisms mediating this segmentation and its coupling to asexual reproductive behavior are still unknown.

The ancestral roles of Hox genes, a family of transcription factors with evolutionarily conserved functions in embryonic head to tail patterning, are tightly linked to the emergence of segmented animal body plans[4,5]. Additionally, Hox gene functions in neuronal development throughout taxa suggest potentially ancient roles in the evolution of the nervous system[6–8]. As asexually reproducing animals, planarians preserve many embryonic developmental programs within their adult tissues, including the ability to establish and

**Fig. 1 Planarian Hox genes are required for adult asexual reproduction. a** Diagram of the patterning and behavioral aspects of planarian asexual reproduction. **b** Fission activity following RNAi against Hox genes. Plot depicts fission progeny number on day 12 of primary RNAi screen ($n = 54$, 12, and 18 animals for Control#1, Control#2, and all other RNAi samples, respectively; $p$-value calculated by Welch's two-tailed $t$-test versus corresponding control; Control#1 vs. *lox5a*, *post2d*, *hox4b*, *post2c*, *lox4*, *hox4a*, *post2a*, *hox3b*, *hox3a*, *hox1*, *lox5b*, *post2b* $p$-values = 0.1733, 0.5958, >0.9999, 0.8582, 0.4191, 0.0455, 0.0348, <0.0001, <0.0001, <0.0001, <0.0001, and <0.0001, respectively, Control#2 vs. *post1* $p$-value = 0.0384). Box and whisker plot depicts individual data points, median (center), 25th/75th percentile (bounds of box), and minima/maxima (whiskers). Source data are provided as a Source Data file. **c, d** Heatmaps depicting fission activity following RNAi treatment for both the (**c**) 2-phase primary and (**d**) secondary Hox RNAi screens. Cumulative fissions over time are displayed for individual worms from each RNAi condition ($n = 12$ or 18 animals). Source data are provided as a Source Data file. **e, f** Representative images of animals from secondary Hox RNAi screen on days 0 and 12 of the fission assay ($n = 12$ animals, three independent repeats). Animals were given nine dsRNA feedings for primary phase I and 12 dsRNA feedings for phase II screening. Animals were given 17 bacterial RNAi feedings for secondary screening. **g** Representative images and phenotypic frequency of RNAi animals 15 days post-amputation after eight dsRNA feedings ($n = 18–53$ animals, three independent repeats). Scale, 1 mm.

maintain the polarization and patterning of the body plan along the A/P axis[9,10]. We therefore hypothesized that Hox genes play a role in A/P directed tissue segmentation and/or transverse fission behavior underlying asexual reproduction in adult planarians.

There has been a great deal of interest in understanding Hox gene function in planaria. Planarian Hox genes were first cloned over 30 years ago[11,12]. Since then, multiple studies have characterized the spatio-temporal regulation of planarian Hox genes during homeostasis and regeneration[11,13–15]. With the introduction of RNA-mediated genetic interference (RNAi)[16], many investigators made extensive efforts to perturb planarian Hox gene function (both individually and combinatorially) with no discernible phenotypic defects reported[15,17,18]. Only recently has a requirement for the planarian Hox gene *post2d* in proper tail regeneration been reported[19]. Thus, to date, the functions of Hox genes in planaria remain largely a mystery.

In this work, we report functions for Hox genes in the asexual reproduction of the planarian flatworm, *S. mediterranea*. We identify multiple planarian Hox genes required for adult tissue segmentation and behavior underlying transverse fission. Among these, RNAi-mediated silencing of *hox3* genes increases both fission segment number and fission behavior initiation. In contrast, RNAi of *post2b* eliminates both fission segmentation and fission behavior. Using RNAseq expression analysis and RNAi functional screening, we identify downstream effector genes mediating Hox gene function in asexual reproduction. We find that *post2b* regulation of effector genes is required for the function of the planarian adhesive organ in the anchoring of the body form to a substrate to initiate transverse fission. Our data support a model in which the coordinate functions of Hox genes regulate asexual reproductive segmentation and behavior within adult planaria.

## Results

### Planarian Hox genes are required for asexual reproduction.
The *S. mediterranea* genome encodes 13 Hox genes: *Hox1, Hox3a, Hox3b, Hox4a, Hox4b, Lox4, Lox5a, Lox5b, Post1, Post2a, Post2b, Post2c, Post2d*[15] (Supplementary Fig. 1a, b). In planaria, Hox genes are dispersed throughout the genome in an atomization of the ancestral cluster[4] (Supplementary Fig. 1c). This genomic organization may have implications for Hox gene expression and function in planaria as Hox genes typically display spatial collinearity (correspondence between gene order along the chromosome with domains of expression along the head to tail axis) in other animals[20]. RNAseq analysis of sexual and asexual planarian biotypes indicates that Hox gene expression increases from embryogenesis to adulthood (10/13 genes) with comparatively minor changes during whole animal regeneration/remodeling[21,22] (Supplementary Figs. 2, 3). Despite an absence of genomic clustering, whole-mount in situ hybridization (WISH) reveals both axially restricted (*hox3b, hox4b, lox5a, post2c,* and *post2d*) and radially layered (*post2a* and *post2b*) domains of Hox gene expression within adult stage planarians[15]. Notably, *hox3b* is enriched in a band of expression anterior to the pharynx in 4–5 mm animals[15]. This region overlaps with the location of newly arisen fission planes as planarians grow from 3 to 5 mm in length[1]. Additionally, *hox1, hox3a, hox4b, post2d,* and *post2b* exhibit overlapping expression with the cholinergic neuron marker, *chat* in the planarian brain and nerve cord[15]. The expression of planarian Hox genes within these axially restricted domains and neuronal cells are consistent with potential functions in the tissue segmentation and behavior underlying asexual reproduction.

We determined the role of each of the planarian Hox genes in asexual reproduction. Individual RNAi of 5/13 Hox genes (*hox1, hox3a, hox3b, lox5b,* and *post2b*) in adult stage planaria resulted in a >50% reduction in the generation of fission progeny

(Fig. 1b, c, Supplementary Fig. 4a–c). Targeting *hox1, hox3a, hox3b, lox5b,* and *post2b* with an extended RNAi treatment in a secondary screen either entirely or largely eliminated fission activity (Fig. 1d, f, Supplementary Fig. 4d). Consistent with previous reports, knockdown of *hox1, lox5b, post2b, hox3a,* or *hox3b* did not result in evident morphological changes during growth, homeostasis, or regeneration[15,17,18] (Fig. 1e–g). These results identify a set of planarian Hox genes with prominent roles in asexual reproduction.

### Role of Hox genes in adult tissue segmentation.
We hypothesized that planarian Hox genes regulate the adult tissue segmentation underlying asexual reproduction. We utilized our previously established physical compression assay to reveal the fission planes dividing tissue segments in RNAi animals and thereby determine the respective effects of Hox gene silencing[1]. Knockdown of *hox1* or *lox5b* had little to no effect on segment number (Fig. 2a, b). In contrast, knockdown of *post2b* eliminated fission segmentation, while knockdown of *hox3a* or *hox3b* resulted in supernumerary segments (Fig. 2a, b). Although simultaneous knockdown of *hox3a + hox3b* increased fission segment number, it also reduced fission progeny yield (Fig. 2a, b, Supplementary Fig. 4e–g). The increased fission segmentation following *hox3* RNAi was slightly biased in the region anterior of the pharynx, coinciding with the expression domain of *hox3b*[15] (Fig. 2c). We reasoned that increasing the number of fission planes dividing the animal should reduce the size of each segment and the resultant fission progeny. We quantified the size of rare fission progeny in *hox3* RNAi animals, revealing a decrease in progeny size consistent with the increase in segmentation observed via physical compression (Fig. 2d, e). Altogether, these data indicate that the planarian Hox genes *post2b* and *hox3* are required for A/P tissue segmentation associated with asexual reproduction.

### Role of Hox genes in asexual reproductive behavior.
We next determined the roles of planarian Hox genes in the frequency, duration, and success of asexual reproductive behavior. Using time-lapse imaging, we recorded fission events in Hox RNAi-treated animals[1] (Fig. 3a, Supplementary Figs. 5, 6, Supplementary Movies 1–8). Individual knockdown of either *hox1* or *lox5b* substantially reduced the success of fission behavior (Supplementary Fig. 6b–f, I, Supplementary Movies 2, 5). Mirroring their opposing roles in segmentation, *post2b* RNAi eliminated fission behavior (Supplementary Figs. 6j, 7b) while *hox3a + hox3b* RNAi increased fission behavior frequency and duration (Fig. 3b–d, Supplementary Figs. 6g h, 7a, c, Supplementary Movies 6, 7). Individual RNAi of *hox3b*, but not *hox3a*, largely phenocopied *hox3a + hox3b* RNAi effects on fission behavior (Supplementary Figs. 6b–d, g, h, 7c, Supplementary Movies 3, 4). These results suggest unequal levels of functional redundancy for the *hox3* genes in the regulation of fission behavior in contrast to their similar contributions to fission segmentation (Fig. 2). Importantly, these results indicate that *hox1* and *lox5b* regulate fission behavior independent of segmentation while *post2b* and *hox3* mediate opposing roles in fission behavior regulation that parallels their roles in segmentation.

Our analysis indicates that planarian *hox3* genes have shared roles in the negative regulation of adult tissue segmentation and asexual reproductive behavior. Yet, while *hox3* RNAi increases the frequency and duration of fission behavior, these attempts are ultimately unsuccessful (Fig. 3b–e). Therefore, we hypothesized that *hox3* is also required at a later stage of the asexual reproduction process. Detailed analyses of fission behavior in control and *hox3* RNAi-treated animals revealed two distinct

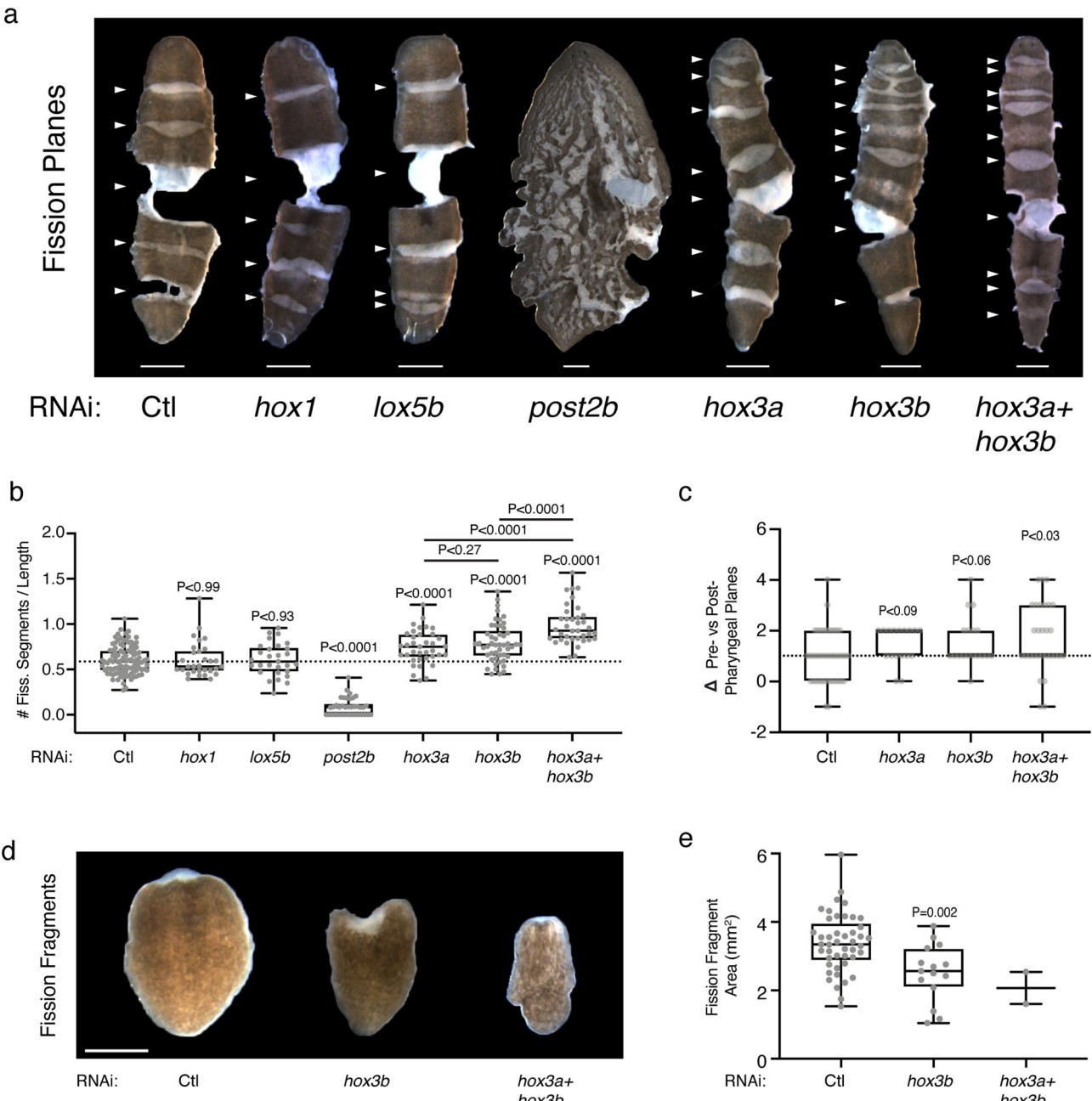

**Fig. 2 *post2b* and *hox3* are opposing regulators of adult tissue segmentation. a** Representative images of RNAi animals compressed to reveal fission planes and segmentation (arrows indicate fission planes). **b** Plot of the number of fission planes/animal length following Hox RNAi treatment ($n = 100$, 30, 29, 38, 37 49, and 37 animals for Ctl, *hox1*, *lox5b*, *post2b*, *hox3a*, *hox3b*, and *hox3a* + *hox3b* RNAi samples, respectively; Ctl vs. *hox1*, *lox5b*, *post2b*, *hox3a*, *hox3b*, and *hox3a* + *hox3b* p-values = 0.9844, 0.9234, <0.0001, <0.0001, <0.0001, and <0.0001, respectively; *hox3a* vs. *hox3b* and *hox3a* + *hox3b* p-values = 0.2696 and <0.0001, respectively; *hox3b* vs. *hox3a* + *hox3b* p-value is <0.0001; sample replicates pooled from two independent experiments with eight dsRNA feedings or 17 bacterial RNAi feedings; at least two independent repeats for each RNAi conditions). **c** Plot of the difference between the number of pre-pharyngeal vs. post-pharyngeal fission planes following RNAi treatment ($n = 41$, 17, 18, and 29 animals for Ctl, *hox3a*, *hox3b*, and *hox3a* + *hox3b* RNAi samples, respectively; Ctl vs. *hox3a*, *hox3b*, and *hox3a* + *hox3b* p-values = 0.0876, 0.0568, 0.0257; 17 bacterial RNAi feedings). **d** Representative images of fission progeny following RNAi treatment. **e** Plot of surface area of fission progeny ($n = 45$, 15, and 2 fission progeny for Ctl, *hox3b*, and *hox3ab* RNAi samples, respectively; Ctl vs. *hox3b* p-value = 0.002; 17 bacterial RNAi feedings). p-values were calculated by Welch's two-tailed t-test. Scale, 1 mm. Box and whisker plot depicts individual data points, median (center), 25th/75th percentile (bounds of box), and minima/maxima (whiskers). Source data are provided as a Source Data file.

phases of behavior (I and II) common to individual fission attempts (Fig. 3f, g). During phase I, animals elongate their bodies anteriorly in a contorted manner. During phase II, animals further stretch out, thinning and rupturing the fission plane dividing segments to release a tissue fragment. Analysis of *hox3*

RNAi-treated animals reveals an inability to progress from phases I to II, thus preventing completion of asexual reproduction (Fig. 3h–j). Furthermore, high-resolution time-lapse imaging of *hox3a* + *hox3b* and *hox3b* RNAi-treated animals revealed that phase I contortions are coincident with A/P directed peristaltic

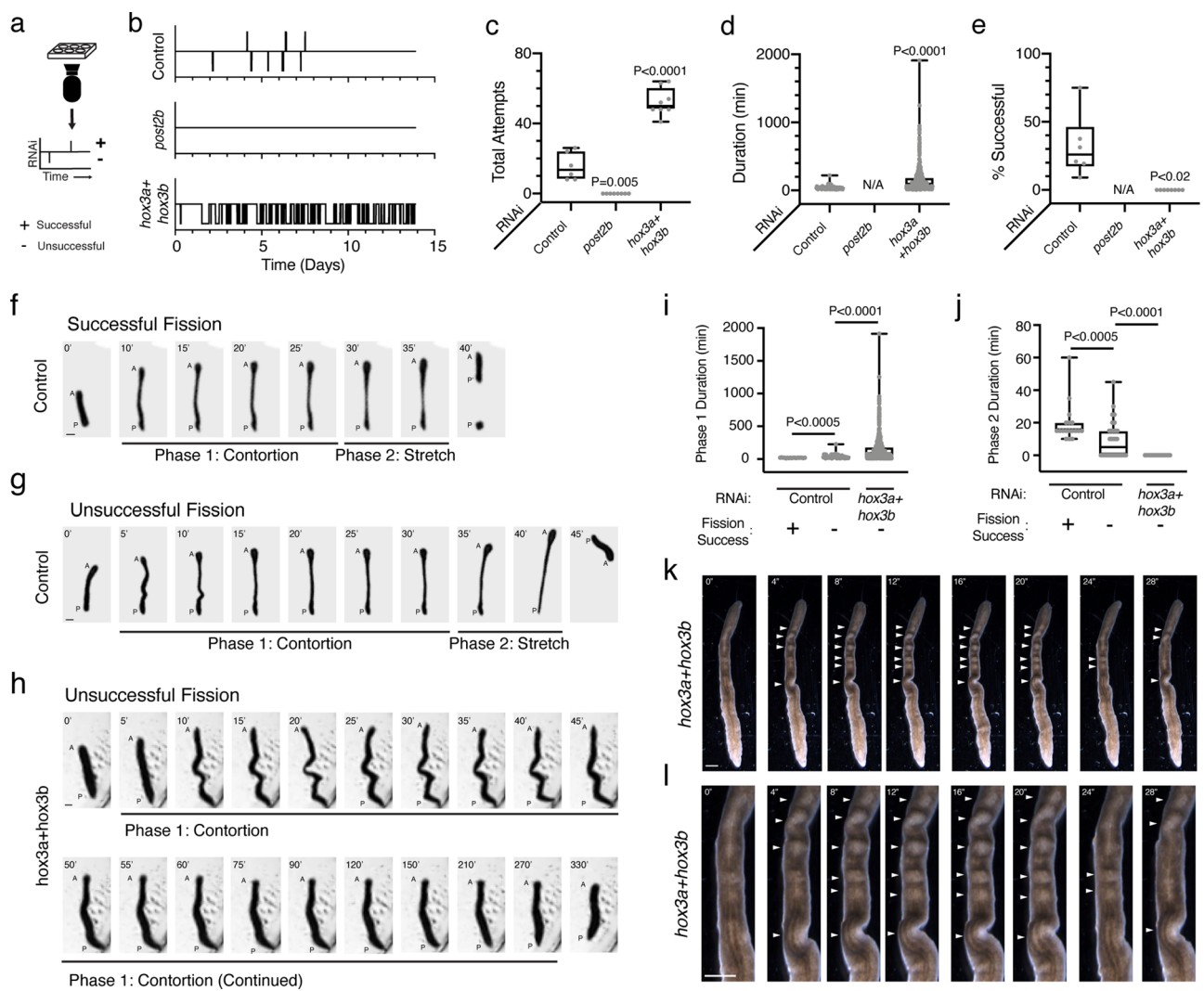

**Fig. 3 post2b and hox3 are opposing regulators of asexual reproductive behavior. a** Diagram of webcam time-lapse imaging and timeline visualization depicting successful (upward displacement) and unsuccessful (downward displacement) fission attempts. **b** Representative activity timelines of fission behavior from live imaging of control, *post2b*, and *hox3a + hox3b* RNAi animals. **c–e** Plots of (**c**) total number of fission attempts (Control vs. *post2b* and *hox3a + hox3b* p-values = 0.005 and <0.0001, respectively), (**d**) duration of each attempt (Control vs. *hox3a + hox3b*, p-value < 0.0001), and (**e**) percentage of successful attempts for RNAi-treated animals (Control vs. *hox3a + hox3b* p-value = 0.194). n = 8, 6, and 6 animals for Control, *post2b*, and *hox3a + hox3b* RNAi samples, two independent repeats. Source data are provided as a Source Data file. **f–h** Representative images (5 min per frame, A = anterior, P = pre-fission posterior, and P′ = post-fission posterior) of (**f**) successful and (**g, h**) unsuccessful fission attempts divided into Phase 1 (Contortion) and Phase 2 (Stretch) for (**f, g**) control and (**h**) *hox3a + hox3b* RNAi-treated animals (experiment independently repeated once). **i, j** Plots of the duration of (**i**) Phase and (**j**) Phase 2 within each fission attempt for RNAi-treated animals (n = 18, 50, and 476 fission events for Ctl Successful, Ctl Unsuccessful, and *hox3a + hox3b* Unsuccessful, respectively; Ctl Successful vs. Ctl Unsuccessful p-value = 0.0005; Ctl Unsuccessful vs. *hox3a + hox3b* Unsuccessful p-value < 0.0001). Source data are provided as a Source Data file. **k, l** Representative images of the (**k**) entire body or the (**l**) pre-pharyngeal region of *hox3a + hox3b* RNAi-treated animal stalled in Phase 1 (n = 10/10 animals, arrows indicate tissues undergoing peristaltic contractions, two independent repeats). Animals were given 17 bacterial RNAi feedings. Scale, 1 mm. Box and whisker plot depicts individual data points, median (center), 25th/75th percentile (bounds of box), and minima/maxima (whiskers). p-values calculated by Welch's two-tailed *t*-test. Scale, 1 mm.

contractions (Fig. 3k, l, Supplementary Movies 9, 10). This peristalsis was previously observed in live imaging of fissioning *S. mediterranea*, and a similar process was observed in fissioning *Dugesia japonica*[1,2]. To date, the significance of these peristaltic contractions in planarian asexual reproduction is still unclear. We conclude from these data that *hox3* is required for both the initiation and successful progression of asexual reproductive behavior.

**Resolving effector genes mediating Hox gene function**. Our findings indicate that Hox genes regulate asexual reproduction via

the emergence and modulation of A/P segmentation and behavior in adult animals. We previously identified requirements for the Wnt signaling genes *β-catenin, apc, dshB*, and *wnt11-6* as well as the TGF-β signaling genes *actR-1* and *smad2/3* in the regulation of fission behavior[1]. Their functions coincided with their roles in the size-dependent patterning of putative mechanosensory neurons expressing *gabrg3L-2* and *pkd1L-2*, genes with roles in the inhibition of fission activity[1,23]. The interplay between Hox and Wnt signaling is an ancient feature of axial patterning broadly conserved throughout phyla[19,24,25]. Additionally, TGF-β and/or BMP signaling has been shown to regulate Hox gene expression and collaborate with HOX proteins for target gene regulation[26,27].

This raises the question as to the extent to which *hox1*, *hox3a*, *hox3b*, *lox5b*, and/or *post2b* interact with Wnt/TGF-β signaling in the regulation of asexual reproduction.

We explored the potential interplay between Wnt/TGF-β genes and the identified fission regulatory Hox genes in planarian asexual reproduction. While knockdown of *β-catenin* alters expression of multiple Hox genes, including *hox4b* (aka *hoxD*), *lox5a*, *post2c*, and *post2d* (aka *abdBa*), there is no significant effect on the expression of *hox1*, *lox5b*, *hox3a*, *hox3b*, or *post2b*[19,28]. Additionally, while the expression domain of *post2b* along the lateral edge of the animal resembles that of *gabrg3L-2* and *pkd1L-2*, single-cell RNAseq (scRNAseq) analysis indicates that they are expressed within non-overlapping cell clusters[15,23,29] (Supplementary Fig. 7e–g). Finally, the RNAi phenotypes of Wnt/TGF-β pathway genes are distinct from that of the identified fission-regulatory Hox genes. Animals treated with *hox1*, *hox3a*, *hox3b*, *lox5b*, or *post2b* RNAi lack detectable anterior–posterior regeneration phenotypes (Fig. 1e, g) and knockdown of *hox3a*, *hox3b*, or *post2b* alters fission segmentation (Fig. 2). In contrast, knockdown of fission regulatory Wnt/TGF-β genes yields anterior–posterior regeneration defects and has little to no effect on fission segmentation[1,28,30–33]. The lack of an obvious functional integration of fission-regulatory Hox genes into the Wnt/TGF-β pathway motivated a de novo investigation of the mechanism by which Hox genes regulate asexual reproduction.

Hox genes mediate their functions via direct and indirect regulation of downstream effector genes[34]. To elucidate Hox downstream effector genes regulating asexual reproduction, we devised a two-part strategy to (I) identify putative candidates via differential gene expression analysis and (II) functionally validate their roles in fission (Fig. 4a). RNAseq analysis identified 1724 DEGs following perturbation of Hox gene function (Fig. 4b, Supplementary Data File 1). We selected and successfully cloned 423 putative effector genes into RNAi vectors for functional analysis. We determined the role of each gene in asexual reproduction by scoring resultant fission progeny following RNAi-mediated gene perturbation (Supplementary Fig. 8a, b). We focused on RNAi conditions that phenocopied Hox gene RNAi—i.e., robustly decreased fission progeny (>4-fold reduction, *p*-value < 0.01) independent of evident alterations to body plan morphology or size (Fig. 1e–g, Supplementary Fig. 4d). Using these criteria, our functional screen identified 24 putative Hox downstream effector genes required for asexual reproduction: *ganglioside GM2 activator* (*gm2a*), *zinc finger protein 816* (*znf816*), *sterile alpha motif-containing protein* (*samc*), *sex determining region Y-box 5* (*sox5*), *carbonic anhydrase VII* (*ca7*), *thymine-DNA glycosylase* (*tdg*), *dd25334*, *G elongation factor mitochondrial 2* (*gfm2*), *C-terminal-binding protein 2* (*ctbp2*), *fukutin* (*fktn*), *dd13286*, *plasminogen-1* (*plg1*), *synaptotagmin 1* (*syt1*), *intermediate filament b* (*ifb*), *lamin A/C* (*lmnAC*), *dd20390*, *post2a*, *Fer3-like bHLH transcription factor* (*fer3l2*), *BarH-like homeobox 2* (*barHl2*), *Nipped-B-like protein 1* (*nipbl*), *zinc finger MYM-type 2* (*zmym2*), *reticulocalbin-1* (*rcn-1*), *even-skipped homeobox 1* (*eve*), *neural cell adhesion molecule 2* (*ncam2*) (Fig. 4c, d, Supplementary Fig. 8c–e).

The uncovered effector genes include putative mediators of *hox1*, *lox5b*, *hox3*, *hox3b*, *hox3a*, and *post2b* gene function in asexual reproduction (Supplementary Figs. 4c, 8c). Of note, we identified *post2a* as a downstream effector gene of *post2b*. Effects of *post2a* RNAi in our primary screen were marginal, but an extension of RNAi knockdown (8X vs. 17X RNAi feedings) increased phenotypic penetrance, substantially reducing fission progeny (Figs. 1b, 4c). Of the remaining effector genes, 20/23 have homology to human genes, indicating that planarian Hox genes regulate asexual reproduction by controlling conserved homologous genes rather than planarian-specific genes.

Furthermore, identifying a Hox downstream effector gene homologous to *ncam2* is noteworthy as N-CAM was the first identified mammalian Hox gene target[35]. Analysis of the 1000 bp upstream regulatory region reveals the presence of conserved *Hox*-binding motifs (TTWATKA) in 22/24 effector genes, including *ncam2* (Supplementary Data File 2). These findings suggest that planarian asexual reproduction may be mediated by conserved Hox gene/target gene regulatory interactions and illustrate the potential of this experimental paradigm for resolving conserved mechanisms of Hox gene function.

**Expression and function of *post2b* effector genes.** To gain insight into the mechanism of action by which Hox genes regulate asexual reproduction, we set out to determine the spatial and cellular context of Hox genes and their effectors. In situ hybridization analysis of Hox gene transcripts poses a significant challenge due to their low expression in planarian tissues[15]. We therefore utilized published scRNAseq expression data to resolve the cell populations in which Hox genes were co-expressed with their cognate downstream effectors[29]. This analysis identified overlapping expression for *post2b* and 11/15 of its downstream effectors (Supplementary Fig. 9a–e). *Post2b* is enriched in the single-cell clusters *smedwi*+ PP, non-ciliated neurons 0, parenchymal 6, 8, 10, 13, and epidermal 10 (Supplementary Fig. 9a–e). Consistent with this analysis, published WISH staining confirms *post2b* expression in a subset of *smedwi*+ cells, the CNS, and layers of parenchymal and epidermal cells along the lateral edge of the animal[15,18,36]. Amongst the 11 effector genes co-enriched with *post2b*, *rcn-1* was expressed in *smedwi*+ PP, and parenchymal-8/13 cell clusters (Fig. 4e, Supplementary Fig. 9a–e, Supplementary Data File 2). The remaining 10 effectors were co-enriched in the parenchymal and epidermal cell clusters: *syt-1* and *fer3l2* in parenchymal 8/13, *eve* in parenchymal 6/10, and *plg1*, *ifb*, *lmnAC*, *post2a*, *barHl2*, *dd13286*, and *dd20390* in epidermal 10 (Supplementary Fig. 9a–e, Supplementary Data File 2). WISH analysis delineated an inner layer of *rcn-1*+ *syt1*+ cells and an outer layer of *lmnAC*+ and *ifb*+ cells along the lateral edge of the animals that paralleled publish *post2b* expression patterns (Fig. 4e, Supplementary Fig. 9f). These scRNAseq and in situ analyses indicate that *post2b* and its respective downstream effector genes are co-expressed within specific planarian cell populations.

We next set out to determine whether *post2b* regulated the expression of these cognate downstream effectors within the identified cell populations. We used WISH to analyze gene expression in *post2b* RNAi-treated worms. Knockdown of *post2b*, but not *hox3*, eliminated lateral edge staining of *rcn-1*, *syt1*, *lmnAC*, and *ifb* (Fig. 4f). These results are consistent with previous reports that *post2b* regulates *ifb* expression in planarian lateral epidermal cells[18]. Notably, RNAi targeting of *post2b* did not affect *rcn-1* expression in parapharyngeal cells outside the domain of *post2b* expression[15]. Collectively, this expression analysis indicates that *post2b* maintains the expression of genes within lateral parenchymal and epidermal cell populations that are required for asexual reproduction.

To gain insight into the mechanism of action of these *post2b* effectors, we determined whether they were required for segmentation similar to *post2b*. Surprisingly, RNAi of tested *post2b* effectors did not phenocopy the loss of segmentation observed in *post2b* RNAi animals (Fig. 4g). We reasoned that these effectors were instead required for fission behavior. Knockdown of *rcn-1*, *plg1*, *syt1*, *ifb*, *lmnAC*, or *post2a* significantly reduced or eliminated fission attempts, largely phenocopying *post2b* RNAi (Fig. 4h, Supplementary Fig. 10, Fig. 3b–e, Supplementary Movies 11–17). RNAi phenotypes of most

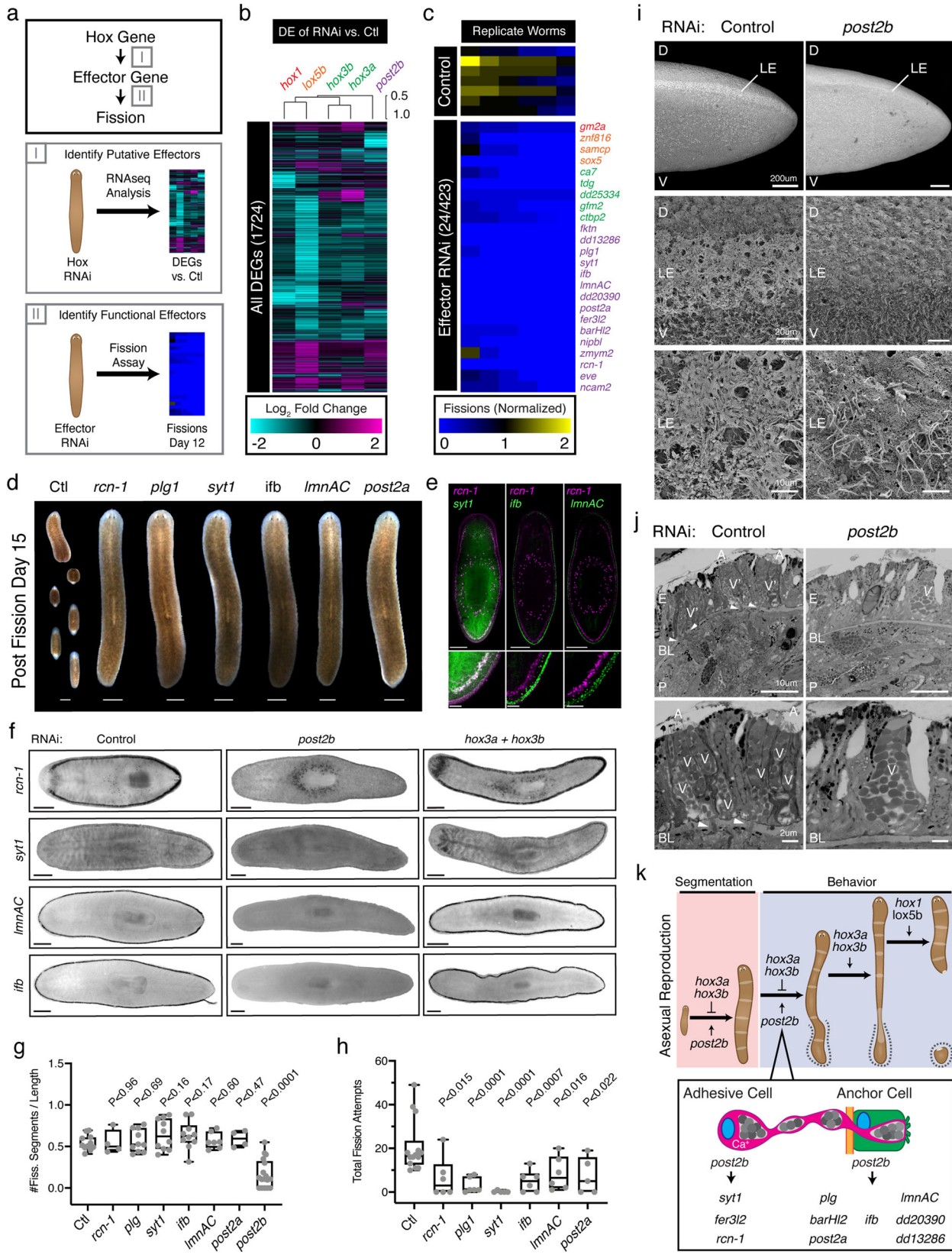

effectors were not as strong as that of *post2b*, suggesting that multiple effectors work in concert to give rise to the *post2b* phenotype. These findings indicate that *post2b*-mediated regulation of effector gene expression within these lateral parenchymal and epidermal cell populations is required for fission behavior.

**Role of *post2b* and effectors in the MAG.** Our screening analysis identified multiple Hox gene effectors required for asexual reproduction (Fig. 4). ScRNAseq analysis and WISH staining indicate that the effectors required for fission are expressed within multiple cell populations in the adult animal[29] (Fig. 4,

**Fig. 4 Elucidation of downstream effectors mediating Hox gene regulation of asexual reproduction. a** Diagram depicting a two-part strategy to elucidate downstream effectors of Hox genes that regulate fission: (I) use RNAseq analysis to identify putative effector genes from the differentially expressed genes in Hox RNAi-treated animals, then, (II) use RNAi to determine the effects of putative effector genes on fission. **b** Heatmap of 1724 DEGs from RNAseq that are significantly regulated following RNAi of Hox genes. **c** Heatmap of the top 24/423 genes from RNAi screen of putative downstream fission effectors (see Supplementary Fig. 8) 16–19 Bacterial RNAi feedings. Heatmap depicts normalized fission number on Day 12 across six RNAi-treated worms. Effector genes are color coded with respect to their putative upstream Hox gene: *hox1* (red), *lox5b* (orange), *hox3* (green), *post2b* (purple). Source data are provided as a Source Data file. **d** Representative images of *post2b* effector RNAi-treated animals 15 days post fission induction ($n = 6$ animals, 17 RNAi feedings, two independent repeats). Scale, 1 mm. **e** Double fluorescent whole-mount in situ staining of *post2b* effectors *rcn-1* and *syt1*, *ifb*, or *lmnAC* ($n = 6$–7 animals, two independent repeats). Whole animal image (top) and zoom in of posterior lateral section (bottom). Scale, 500 and 100 μm, respectively. **f** Expression of *post2b* effector genes detected by whole-mount in situ hybridization in control, *hox3a + hox3b*, and *post2b* RNAi-treated animals ($n = 4$–13 animals, 1 independent repeat). Scale, 500 μm. **g** Plot of the number of fission planes/animal length following *post2b* effector RNAi treatment ($n = 4$, 10, 11, 10, 10, 7, 4, and 17 animals for Ctl, *rcn-1*, *plg*, *syt1*, *ifb*, *lmnAC*, *post2a*, and *post2b*, respectively; Ctl vs. *rcn-1*, *plg*, *syt1*, *ifb*, *lmnAC*, *post2a*, and *post2b* p-values = 0.9581, 0.6815, 0.1552, 0.1634, 0.5967, 0.4605, and 0.0001, respectively; replicates pooled from 1 to 4 independent repeats, 21 bacterial RNAi feedings). p-Values were calculated by Welch's two-tailed t-test. Source data are provided as a Source Data file. **h** Plot of the total number of fission attempts for RNAi-treated animals ($n = 14$, 5, 5, and 6 animals for Ctl, *syt1*, *post2a*, and all other RNAi samples, respectively; Ctl vs. *rcn-1*, *plg*, *syt1*, *ifb*, *lmnAC*, *post2a*, and *post2b* p-values = 0.0145, <0.0001, <0.0001, 0.0006, 0.0157, and 0.0218, respectively; 21 bacterial RNAi feedings. Box and whisker plot depicts individual data points, median (center), 25th/75th percentile (bounds of box), and minima/maxima (whiskers). p-values were calculated by Welch's two-tailed t-test. Source data are provided as a Source Data file. **i** Scanning electron micrographs of the lateral edge of the tail containing the marginal adhesive gland in control and *post2b* RNAi-treated animals ($n = 3$ animals; representative images; 30 bacterial RNAi feedings; two independent repeats). **j** Transverse scanning electron micrographs of the marginal adhesive gland in control and *post2b* RNAi-treated animals ($n = 3$ animals; representative images; 30 bacterial RNAi feedings; single experiment). Annotations demarcate the dorsal (D), ventral (V), lateral edge (LE), epithelium (E), basal lamina (BL), parenchyma (P), viscid cell extensions crossing basement lamina (arrowheads), clustered (V') or individual (V) viscid cell glands in the epidermis, and adhesive papillae (A). **k** Diagram summarizing the roles of the Hox genes as regulators of the adult tissue patterning and behavior required for asexual reproduction (top). Zoom-in panel describing requirement of *post2b* in the initiation of fission behavior via effector gene regulation in the adhesive cells and anchor cells of the planarian marginal adhesive glands (bottom).

Supplementary Fig. 9). These observations suggest that gene function within multiple tissues and cell types, in addition to the CNS, are required for asexual reproduction and even different aspects of fission behavior[1]. But, in the case of the identified *post2b* effectors, how are gene functions within lateral parenchymal and epidermal cells required for fission behavior initiation?

Cross-referencing scRNASeq data, in situ hybridization data, and published literature revealed that the lateral parenchymal and epidermal cell populations are *mag-1+* viscid gland cells and *ifb+* epithelial anchor cells of the marginal adhesive gland (MAG)[18,37]. Within this duo gland organ, conserved across platyhelminthes, the coordinate functions of adhesive gland cells (viscid gland cells), releasing cells, and anchor cells mediate transient adhesion to substrates[38,39]. Viscid gland cells within the parenchyme expel their adhesive secretions onto the animal's surface via long cellular processes that cross the basal lamina and travel through epithelial anchor cells to outer pores surrounded by adhesive papilla[38,39]. In *Macrostomum lignano*, RNAi targeting of the anchor cell-specific intermediate filament gene *macif1* results in severe morphological alterations in anchor cells and an inability to adhere to substrates[39]. Assuming conservation of gene functions, RNAi targeting of the anchor cell-specific planarian intermediate filament gene *ifb* likely compromises the structure and function of the epithelial secretory pore. Furthermore, the *post2b* effector *syt1* is expressed in viscid gland cells and is homologous to *synaptotagmin*, a calcium sensor for regulated exocytosis[40]. Additionally, the *post2b* effect *rcn-1*, also expressed in viscid gland cells, is homologous to *reticulocalbin*, a calcium-binding protein of the secretory pathway[41]. Based on the functions of these homologous genes, we predict that RNAi targeting of *rcn-1* and *syt1* likely compromises calcium-regulated exocytosis of adhesive secretions from the viscid gland. Notably, RNAi targeting *post2b* or downstream effectors expressed within the marginal adhesive organ (*rcn-1*, *plg1*, *syt1*, *ifb*, *lmnAC*, and *post2a*) results in animals constantly drifting across substrates while adopting a resting position (Supplementary Movies 7, 12–17). This phenotype is consistent with a defect in substrate adherence and inability to stop cilia-driven locomotion and remain stationary. Given that fission behavior begins with the anchoring of the worm's posterior end to a substrate and requires functional effectors specifically expressed within the MAG, we conclude that control of MAG constitutes a key facet of planarian asexual reproduction.

To investigate the function of *post2b* in the MAG, we used electron microscopy to characterize the cellular structure of the viscid gland and adhesive cells in *post2b* RNAi animals. Scanning electron micrographs (SEM) of an amputated planarian tail region revealed a distinct mucus stripe along the lateral edge epithelial surface, corresponding to the adhesive secretions of the MAG (Fig. 4i, Supplementary Fig. 11a). Knockdown of *post2b* eliminated or vastly reduced this stripe of mucosal secretions. Notably, the exposed lateral edge epithelial surface of *post2b* RNAi animals had expected complements of cilia and microvilli but lacked detectable adhesive papillae. Analysis of transverse sections in control animals resolved parenchymal viscid gland cells extending processes that crossed the basal lamina to connect to adhesive papillae-covered pores on the epithelial anchor cell surface (Fig. 4j, Supplementary Fig. 11b). Consistent with our exterior SEM analysis, *post2b* RNAi animals lacked a distinct anchor cell region in the lateral margin epithelium and displayed a reduced number of viscid gland cell extensions crossing the basal lamina and terminating at the epithelial surface. Of the few viscid gland cells detected, all lacked adhesive papillae around the pore termini. Altogether, our findings indicate that *post2b* functions in the MAG are required for initiation of planarian fission behavior.

## Discussion

Our study identifies roles for planarian Hox genes as mediators of adult tissue segmentation and fission behavior in addition to providing evidence that Hox genes function in asexual reproduction. In combination with prior work, our current findings indicate complementary roles for Hox and Wnt/TGF-β genes during planarian asexual reproduction[1]. The coordinate roles of *hox1*, *lox5b*, *hox3a*, *hox3b*, and *post2b* regulate segmentation for progeny tissue

allocation, adhesion for posterior anchoring to substrates, and the execution of transverse fission behavior to ultimately separate tissue fragments that will regenerate into clonal progeny (Fig. 4k). In parallel, Wnt/TGF-β activity coordinates growth with size-dependent patterning of fission-modulatory mechanosensory neurons[1]. Given the preponderance of evidence for the interplay of Hox genes with the Wnt and TGF-β pathways[19,24–27], it would be of interest to further investigate the extent of their interdependence in the regulation of asexual reproduction. Our study also elucidates Hox gene-mediated segmentation within historically designated unsegmented flatworms[42]. In combination with the discovery of Hox-mediated segmentation in the basal metazoan *Nematostella vectensis*, these findings support ancient roles for Hox genes in the regulation of body segmentation in addition to their well-established roles in segmental patterning[43]. Finally, we identify roles for Hox genes in multiple aspects of asexual reproductive behavior, building upon the known roles for Hox genes in developing nervous systems and functions linked with specific animal behaviors[6–8,44]. From cnidarians to deuterostomes, Hox genes play conserved roles in the regulation of axial patterning[4,43]. Notably, asexual reproduction, via budding or fission, is also distributed throughout these groups and phyla[45]. We speculate that the regulation of asexual reproduction and/or key genes underlying the neural circuitry or specific cell types utilized in this process represent ancestral and potentially conserved functions for Hox genes.

## Methods

**Animal husbandry**. Clonal CIW4 strain *S. mediterranea* were maintained in 1× Montjuic salts according to standard husbandry protocols[1]. CIW4 animals were sourced from a large recirculation culture and placed directly into a unidirectional flow system culture for RNAi feeding experiments[1,46]. The planarian recirculation culture system is composed of three culture trays (96′L × 24′W × 12′H) stacked vertically on top of each other over a sump. Planarian water flows from the sump pump through a chiller, canister filter, and a UV sterilizer into the top tray. Water subsequently flows down drains at the opposite ends of the source through the series of three trays. Water then re-enters the sump where it is filtered through two vertically stacked 400 and 200 μm sieves. Beyond the sieves, water is gravity fed and mechanically filtered through a set of filter/floss pads. Finally, water passes through Water Garden Oasis Pond Matrix and Kaldness media to remove nitrogenous waste. Water is then able to flow back through the chiller, canister filter, and UV sterilizer back into the top tray. For the unidirectional flow system, planaria were cultured in disposable plastic cups lined with a pocket of 150 μm nylon mesh (Saatitech, Italy). Cups were perforated with holes at the desired planaria water level for outflow to complete the flow vessel. These flow vessels were placed in an enclosed container on a perforated tray suspended above a collection basin and centralized waste drain. Planarian water was pumped through spigots directly above the flow vessels. The system actively pumped 2 vessel exchanges of water every 12 h. After RNAi feedings uneaten food was removed.

**Gene cloning and RNAi feeding protocol**. Gene cloning was performed as previously described but with the following distinctions[1]. Primer3 version 4.1.0 (https://primer3.ut.ee/) was used to design primer pairs targeting genes of interest, using the Sánchez Alvarado lab transcriptome as a reference. 5′ overhangs were added to ends of both forward and reverse primers to enable downstream Gibson Assembly. Amplified products were designed to be 400–600 base pairs in length. PCR was performed using cDNA as a template. Products were inserted into a modified version of the pPR-T4P vector and transformed directly into *Escherichia coli* strain HT115. Final cloning products were verified by sequencing. A list of all accession numbers and primer sequences for all the Hox genes and effector genes in this study are available in Supplementary Data File 3. Preparation of dsRNA and RNAi feedings were performed as previously described with the following distinctions[1]. Purified dsRNA food was prepared by mixing 1 volume of dsRNA at 400 ng/ml with 1 volume of beef liver paste. Bacterial RNAi food was prepared by mixing 8 ml of pelleted bacteria with 120 μl of beef liver. Food was administered every 3 days. Animals were fed from a starting size of <4 mm to a final size of >10 mm. To achieve this, worms were either fed 7–9 times with purified dsRNA food or 16–19 times with bacterial RNAi food as indicated. Control animals were fed RNAi food targeting *unc-22*.

**Fission assay**. For fission induction, animals were removed from recirculation or unidirectional flow system, rinsed 10 times with planarian water, and individually placed into wells of a six-well tissue culture plate. The number of fission fragments was scored every 1–2 days. For data analysis, the number of daily cumulative fissions was directly plotted or normalized to the average of the control RNAi fissions where indicated. Fission number was converted to a heat color code for visualization.

**Fission segmentation compression assay**. Fission segmentation was revealed by compression of worms between a glass coverslip and plastic tissue culture dish. Animals were inverted with their ventral side up, compressed using four fingertips, then imaged. To ensure that all compression/fission planes were revealed for every animal, images were acquired sequentially using a Leica M205 microscope as each fission plane was revealed by mechanical compression.

**Microscopy**. Images of live worms and fixed colorimetric in situ samples were acquired using a Leica M205 microscope. Fission fragment area was calculated using Adobe Photoshop. Multicolor fluorescent in situ hybridization (FISH) single confocal images were acquired using a Nikon CSU-W1 inverted spinning disk confocal microscope. Fiji plugins were used to stitch tiles for whole planarian images. Following image acquisition, brightness and contrast were adjusted, and the images were rotated and cropped for data presentation.

**Live imaging of fission behavior**. For all data sets, worms were plated in six-well dishes and imaged in one of two ways. For time-lapse A (Supplementary Figs. 5, 6, Supplementary Movies 1–5), worms were imaged from above with a single 4k webcam (Logitech Brio). Four inverted LED ring lights (AmScope) mounted above the camera were used for illumination. Images were acquired using an in-house written Python script described previously[1]. In time-lapses B and C (Figs. 3 and 4, Supplementary Fig. 7, 10, Supplementary Movies 6–17), images were acquired with a configuration wherein the camera was mounted below a large (2 ft × 2 ft) plex-iglass panel that was topped with large white plexiglass panels. The same four LED lights were placed at the bottom of the set-up next to the camera. In this format, a webcam (Logitech c920) was used to acquire images. The time-lapse was acquired using micro-manager version 1.4.22 (https://micro-manager.org/) and the open-cvgrabber framework. In both cases, the camera gain, exposure, and autofocus were controlled using the Logitech Webcam Controller software (https://download01.logi.com/web/ftp/pub/video/lws/lws280.exe).

Videos of individual animals were analyzed, manually annotated, and example frames were extracted using Image J[1]. Fission attempt initiation was scored at the frame in which a rapid stationary elongation occurred following a period of contraction and immobility. Termination of fission attempt was scored at the frame in which the animal ruptured and released a fragment (successful), or the animal returned to its original pre-elongation state (unsuccessful). Fission events were divided into phase I (initial elongation accompanied by body contortions) and phase II (second elongation accompanied by thinning and/or rupture). Data were visualized with timelines plotted using Graphpad Prism version 9.2.0. Worms that crawled out the well and desiccated during the experiment were not considered for further analysis.

Peristaltic activity of *hox3b* and *hox3a + hox3b* RNAi-treated worms was captured using a Leica M205 microscope (acquired at 10 frames per second, Supplementary Movies 9,10).

**RNAseq analysis**. Control or Hox gene RNAi animals (7× purified dsRNA feedings) were starved for 5 days and snap-frozen for RNA collection (4 replicate samples with three animals each). RNA was extracted using Trizol Reagent, purified and Dnase treated (Qiagen RNeasy Mini Kit Cat. No. 74104)[46]. Libraries were prepared according to the manufacturer's instructions using the TruSeq Stranded mRNA Prep Kit (Illumina). The resulting libraries were purified using the Agencourt AMPure XP system (Beckman Coulter) then quantified using a Bioanalyzer (Agilent Technologies) and a Qubit fluorometer (Life Technologies). Libraries were re-quantified, normalized, pooled and sequenced on an Illumina NextSeq 500 instrument as High Output 75 bp single read runs using NextSeq Control Software version 2.2.0.4. Following sequencing, Illumina NextSeq Real-Time Analysis version 2.4.11 and bcl2fastq2 version 2.20 were run to demultiplex reads and generate FASTQ files. These FASTQ files were then aligned to the smed_dd_g4 genome using STAR and parameters: --outFilterMultimapNmax 2 --alignIntronMax 5000 --quantMode GeneCounts, along with the Smes_hv_v1 gene models from the Max Plank Institute, obtained via Planmine, to generate read counts to genes[47]. Differentially expressed genes were determined using the edgeR library in R[48]. *p*-Values were adjusted for multiple hypothesis testing by the method of Benjamini and Hochberg[49]. Cutoff for significant differential gene expression for *hox1*, *hox3a*, *hox3b*, and *post2b* RNAi versus control RNAi was a fold change of 1.5 and an adjusted *p*-value < 0.5. Cutoff for significant differential gene expression for *lox5b* RNAi versus control RNAi was a fold change of 2 and an adjusted *p*-value < 0.1.

**Planarian Hox gene, transcript, and protein sequence analysis**. Planarian Hox gene mRNA and amino acid sequences were obtained from Planosphere (https://planosphere.stowers.org/find/genes). The genomic contig location of planarian Hox genes was determined using PlanMine version 3.0 (http://planmine.mpi-cbg.de/planmine/blast.do). Homeodomains of planarian Hox genes were annotated using Pfam 33.1 (https://pfam.xfam.org/)[50]. Homeodomain sequences were aligned via Clustal Omega (https://www.ebi.ac.uk/Tools/msa/clustalo/).

**Planarian Hox gene expression**. Expression of Hox genes during regeneration and development was obtained from Zeng et al. (2018) and Davies et al. (2017), respectively[21,22]. Single-cell data and t-SNE plots were obtained from a planarian cell transcriptome atlas (https://digiworm.wi.mit.edu/)[29].

**Data visualization and statistical tests**. Heatmaps were generated using Microsoft Excel (version 16.49) and Multiple Experiment Viewer (MeV version 4.8.1). Graphs were generated using GraphPad Prism (version 9.2.0). Illustrations were made using Adobe Illustrator 2020 (version 24.1.3). Images were prepared for figures using ImageJ (version 2.1.0), and Adobe Photoshop 2020 (version 21.1.3). For pair-wise comparisons, significance was calculated using an unpaired $t$-test with Welch's correction using GraphPad Prism (version 9.1.0).

**Whole-mount in situ hybridization**. Colorimetric WISH hybridization was performed as follows[46]. Mucus was removed with 7.5% NAC in PBS for 8 min. Worms were fixed in 4% PFA 0.3% Triton X for 30 min and washed twice with 1× PBS 0.3% Triton-X (PBSTx). Worms were washed twice with PBSTx 0.3%, and then dehydrated with a 50% MeOH:50% PBSTx 0.3% solution, and washed and stored in 100% MeOH O/N at −20 °C. In preparation for sample bleaching, worms were rehydrated with a 50% MeOH:50% PBStX 0.3% wash, washed twice with PBSTx 0.3%, and washed once in 1× SSC. For bleaching, worms were incubated in a 1.2% $H_2O_2$, 5% Formamide, 0.5× SCC solution for 3 h over a light box. Worms were washed once with 1× SSC and twice with 1× PBS 0.3% Tween-20 (PBSTw 0.3%). Worms were permeabilized for 20 min with a 1× PBS 0.1% SDS solution containing 1–2 μg/ml Proteinase K (in accordance with worm size). Worms were post-fixed in 4% PFA 0.3% Tw for 10 min and washed twice with PBSTw 0.3%. Worms were prepared for hybridization with a 10 min 50% solution wash and subsequent 2 h 56 °C incubation with a prehybridization solution containing 50% Formamide, 1× Denhardts solution, 100 μg/ml Heparin, 1% Tween-20, 50 mM DTT, and 1 mg/ml Sigma Torula Yeast RNA in 5× SSC. DIG-labeled probes were denatured 5 min at 70 °C, and incubated O/N at 56 °C in a hybe solution identical to prehybe solution but with Deionized Formamide in place of Formamide, Calbiochem Yeast RNA 0.25 mg/ml in place of Sigma Torula Yeast RNA, and an addition of 5% Dextran Sulfate.

The next day non-specific probe binding was washed out at 56 °C. The samples were washed twice for 30 min in a Wash Hybe solution containing 50% Formamide, 0.5% Tween-20, and 1% Denhardts in 5× SSC. Worms were then washed twice in 50% Wash Hybe:50% 2× SSC 0.1% Tw for 30 min, thrice with 2× SSC 0.1% Tw for 20 min, and thrice with 0.2× SSC 0.1% Tw for 20 min. The samples were returned to room temperature and washed twice with 1× MAB 0.1–0.3% Tw (MABTw) pH 7.5. The samples were blocked for 2 h in a solution of 1× MABTw containing 10% Horse Serum and 0.5% Roche Western Blocking Reagent (RWBR) and stained with anti-DIG and/or anti-DNP Fab fragment conjugated to alkaline phosphatase at a 1:1000 dilution O/N at 4 °C. Non-specific binding was removed with washing six times with MABTw for 2–3 h and then processed for colorimetric development. Worms were incubated for 15 min in solutions of 0.1 M Tris pH 9.5, 0.1 M NaCl, 0.05 M MgCl₂, 0.1% Tween containing 0%, 50%, and then 80% PVA. The final solution of 80% PVA was supplemented with 1:188 BCIP or 1:94 NBT and worms monitored for color development. The reaction was stopped by rinsing 1–2 times in PBS, fixed in 4% PFA 0.3% Tx for 45 min, and washed three times with PBSTx 0.3%. The samples were then washed with 100% EtOH for 20 min and then with 50% EtOH:50% PBS for 5 min. The samples were the rehydrated with 1× PBS washes and mounted in 80% glycerol.

Double FISH was performed for colocalization analysis of Hox effector genes in whole asexual planarian *S. mediterranea*[51,52]. To kill and remove mucus, animals were placed in 5% N-Acetyl Cysteine (NAC, Sigma) in 1× PBS solutions at room temperature with gentle shaking by hand for 5 min. To fix, animals were placed in 4% paraformaldehyde (PFA, Sigma) in 1× PBSTx 0.5% Triton-x at room temperature for 30 min. The reduction solutions step was skipped to preserve the fragile epidermis of the animals. Animals were bleached with 5% non-deionized formamide, 1.2% $H_2O_2$ in 0.5× SSC for 2 h under direct light. Proteinase K treatment (2 μg/ml Proteinase-K, Invitrogen, 0.1% SDS in 1× PBSTx) was performed for 10 min at room temperature, followed by a post-fixation (4% PFA) for 10 min at room temperature. Samples were simultaneously incubated with two riboprobes overnight at 56 °C. Following hybridization, DIG-probes were detected with antibodies (Roche anti-DIG-POD 1:1000) in MABT containing 5% horse serum and 0.5% RWBR. DIG-probes were developed with rhodamine tyramide (1:5000) in Borate Buffer for 45 min. For multicolor FISH, peroxidase activity was killed with 200 mM sodium azide in PBSTw (0.3% Tween). FITC-probes were detected with antibodies (Jackson ImmunoResearch anti-FITC-POD 1:3000) in MABT containing 5% horse serum and 1% RWBR. FITC-probes were developed in FAM tyramide (1:5000) in Borate Buffer for 45 min. DNA was stained with DAPI overnight in MABT. Lastly, FISH animals were cleared overnight in a modified ScaleA2 solution (20% glycerol, 2.5% DABCO, 4 M urea) prior to mounting and imaging.

**Electron microscopy**. For exterior SEM amputated tail fragments were fixed in relaxant/fixative as previously decribed[53], but modified to 3.4% paraformaldehyde, followed by overnight fixation in cold fixative containing 2.5% glutaraldehyde and 2% paraformaldehyde in 50 mM cacodylate buffer (pH 7.35) at 4 °C on a nutator. After waiting 24 h, samples were processed with a TOTO protocol[54], then critical point dried with a Samidri-795 (Tousimis). Tail fragments were mounted on a stub and sputter coated with 4 nm, Gold Palladium with a Leica EM ACE600. Low magnification exterior SEM images were taken on a TM4000Plus (Hitachi) at 15 kV. Higher magnification exterior SEM images and internal SEM images were acquired on a Zeiss Merlin SEM in analytical mode with a Thermo Fisher Scientific/FEI Tecnai G2 Spirit BioTWIN with Gatan UltraScan 1000 CCD camera. High magnification exterior SEM was acquired with the SE2 detector (3 kV, 100 pA). For STEM imaging, animals were processed as previously described[36] and were dehydrated and infiltrated using Hard Plus resin (EMS). Sections (500–100 nm) were cut on a Leica EM UC7 ultramicrotome, transferred to a glass slide and post-stained with Sato's lead and 0.5% uranyl acetate and images with a BSD at 7 kV, 1.2 nA. Both gross brightness and contrast and local contrast were adjusted using CLAHE (blocksize:127, bins: 256, slope:1.5) in Fiji[55].

**Summary of data presentation and statistics**. All data points derived from distinct single measurements of individual biological replicates except for plots of the size of fission fragments or duration of fission attempts wherein multiple distinct data points (not technical replicates) are derived from each biological replicate. $p$-Values calculated with Welch's $t$-test were unpaired, assumed a Gaussian distribution, and did not assume equal standard deviations. Data points were omitted under the following conditions: animal compressions failed due to technical error, animals crawled out of wells and desiccated on lids during the fissioning experiment, and animals recently fissioned prior to recording of animal body length. Details of all statistical analyses performed in Supplementary Data File 4.

**Reporting summary**. Further information on research design is available in the Nature Research Reporting Summary linked to this article.

## Data availability

The original data generated in this study have been deposited in the Stowers Original Data Repository under accession code LIBPB-1567. The Hox RNAi RNAseq data generated from this study has been deposited in NCBI GEO with accession number: GSE159876. The fission progeny scoring data, compression and fission segment scoring data, fission progeny area calculations, fission behavior scoring data, and animal length measurements generated in this study are provided in the Source Data File. The RNAseq data for Supplementary Figs. 2 and 3 used in this study are available in the NCBI GEO database under accession codes GSE82280 and GSE107874. The scRNAseq data used to generate t-SNE plots used in this study is available at https://digiworm.wi.mit.edu/ and NCBI GEO: GSE111764 Source data are provided with this paper.

## Code availabilty

The fission behavior timelapses were acquired using micro-manager version 1.4.22 (https://micro-manager.org/) and the opencvgrabber framework. In both cases, the camera gain, exposure, and autofocus were controlled using the Logitech Webcam Controller software (https://download01.logi.com/web/ftp/pub/video/lws/lws280.exe). Source data are provided with this paper.

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

## Acknowledgements

We thank members of the ASA laboratory for discussion and advice and Robb Krumlauf for their comments. We are grateful to the Stowers Aquatics (particularly the Planarian team), Microscopy, and Molecular Biology core facilities for technical contributions and methods development. A.S.A. is a Howard Hughes Medical Institute (HHMI) and Stowers Institute for Medical Research Investigator. C.P.A. is a Stowers Institute for Medical Research Postdoctoral Fellow. This work was supported in part by NIH R37GM057260 to A.S.A.

## Author contributions

Conceptualization, data analysis, and interpretation (C.P.A., A.S.A.); acquisition of data (C.P.A., A.M.L., F.G.M., J.O.H., S.H.N., J.J.L.); design, fabrication, and software for planarian live imaging systems (J.J.L.); cloning of RNAi vector library for gene perturbation and screening (F.G.M.); RNAseq Differential Gene Expression Analysis (C.W.S.); writing of the original manuscript (C.P.A.); supervision and funding acquisition (A.S.A.); and revision and editing of the manuscript (all authors).

## Competing interests

The authors declare no competing interests.
