## [Peer Review File · Nature Communications]

Hox genes regulate asexual reproductive behavior and tissue segmentation in adult animalsREVIEWER COMMENTS

Reviewer #1 (Remarks to the Author):

This is an interesting follow-up paper of a previous one entitled "The role of Wnt and TGF β signaling in the control of size-dependent behavior of the flatworm *Schmidtea mediterranea*" (Nature 572, 655–659). In that first paper, the authors have described a process of asexual reproduction in planarians during which adult animals stretch and contract their tail tissue and produce posterior tissue fragments that regenerate into intact animals. This process is depending on the size of the animals, it involves active biomechanical forces and it seems to be under the control of a nervous subsystem acting downstream of Wnt/TGF β signaling. In the paper under review the authors analyzed now the role of hox genes in the same process. They identified by gene silencing 5 hox genes that are required for the fission process, the flatworms form of asexual reproduction. Silencing of hox3 and post2b revealed an opposing role of both hox genes: Silencing of hox3 resulted in supernumerary segments, while silencing of post2b inhibited fission completely. Downstream effector genes that mediate hox gene regulation were also identified and they revealed a detailed picture of postembryonic function of hox genes in planarians.

The manuscript is well written and provides a substantial data set, which allows the statement that hox genes are active in the fission process of asexually reproducing planarians. There are two principal comments on the manuscript.

(i) It is certainly interesting to learn more about the function of hox genes in adult organisms, although it is not surprising that hox genes are active in adult planarians. Those are asexually reproducing animals, in which embryonic features are always preserved in adult animals, as they are necessary for the patterning of new structures. My suggestion is to tone down this argument.

(ii) But more importantly, the authors have previously explained fission through the effect of Wnt and TGF signaling on the nervous system and patterning processes. While it's obvious that transcription factors act on a different cellular control level than signaling factors (i.e., Wnt/TGF β signaling), the interesting and important question is now how are the two mechanisms linked. The authors address this point at the end of the manuscript and come to the conclusion that "that a subset of Wnt-independent Hox genes mediate specific functions in asexual reproduction and work in concert with Wnt-dependent patterning and regeneration programs". But what is the proof that these five hox genes and especially hox3 and post2b, are Wnt independent? These important conclusions are not yet sufficiently substantiated. I am also wondering why the authors put much effort in the identification of the target genes of the identified hox genes instead of trying to uncover the interrelationship of Wnt/TGF β signaling and hox genes in this segmentation-like process in planarians. It would be very helpful to have seen some ATACseq data or promotor studies for the five hox genes identified in order to bridge the gap between both papers.

Nevertheless, I support publication of this interesting manuscript.

Reviewer #2 (Remarks to the Author):

This report identifies unexpected functions for Hox genes and their regulatory targets in the process of planarian asexual reproduction by fissioning. In this process, planaria rip themselves apart at specified locations on the AP axis, followed by regeneration to form new progeny. The authors had previously established assays for monitoring animal behavior before and during fission attempts, that unfissioned animals have latent "segments" of compressional mechanical sensitivity marking the sites of future fissioning, and that Wnt signaling and CNS activity are required for controlling the fissioning mechanism.

Here, Arnold et al examine Hox genes for potential roles in this process, finding three categories of defects following Hox gene RNAi. Inhibition of *hox1* and *lox5b* caused overall fission failure with no defects to compression planes ("segments") and instead failure of the behavior of fissioning attempts. Inhibition of *hox3a/b* caused overall fission failure along with excess compression planes and increased frequency of attempts. Finally, inhibition of *post2b*, a laterally expressed hox gene, disrupted compression planes entirely and also eliminated fission behavior. To understand the cellular mechanisms underlying these effects, the authors used RNAseq to identify transcripts dependent on the *hox1*, *lox5b*, *hox3a/b* and *post2b* genes. They inhibited 423 of ~1700 such identified transcripts and found 24 of these were required for fissioning, representing putative downstream targets of each of the initial Hox genes uncovered by the study. Hox-binding motifs could be identified in the regions around most of these targets, suggestive that they could be direct targets. Focusing on *post2b* targets, the authors then cross-referenced with cell atlas data to identify the cell types co-expressing *post2d* and the targets as representing two different putatively radial/lateral parenchymal cell types and a nearby epidermal cell type located at the D/V boundary. None of these factors was important for specifying compression plane segments but were instead important for fissioning behavior. Together the paper makes several important contributions, showing unexpected roles for Hox genes in controlling fission behavior and the planarian segments. The segmentation phenotype of *post2b* RNAi is particularly striking and reveals a new process in the segmentation mechanism. In addition, the analysis to identify targets involved a substantial RNAi screen, revealing novel regulators of the fissioning process. Altogether, the work makes a substantial step forward in understanding the fissioning mechanism which is a critical component of understanding growth and regeneration in these animals.

Comments:

The evidence at hand does not yet definitively show the involvement of the lateral cell populations in mediating fissioning behavior, given the small number of cells recovered by scRNAseq and lack of readily available information in the manuscript for the expression of these target genes. Showing *in situ* hybridizations verifying that lateral/radial expression of factors is dependent on *post2b* would offer significantly greater support for the model in 4g. Similarly, double-FISH experiments would verify the cell-type specific expression detected by drop-seq. As it stands, it is unclear from the data presented how specifically expressed are the targets and whether *post2b* regulation affects this expression.

The paper builds toward uncovering functions for the targets downstream of *post2d* mediating segmentation and fissioning behavior, but surprisingly none is required for generating compression planes (segments). What is the author's interpretation of these results? Is segmentation itself likely controlled by alternate *post2d* expressing cells, factors not included in the screen, or through redundant processes among these factors or by some other process? Based on the evidence from the screens so far, it would seem that fissioning behavior is modified by a larger number of inputs than physical segmentation. Is it possible that the design of the screen scoring could have prevented identification of factors acting downstream of *hox3* or *post2d* specifically in the segmentation process?

The data showing lack of evidence for an effect of inhibiting *post2d* downstream genes on segmentation needs to be shown because of the central role this plays in the model. It is also unclear as stated, exactly what experiment was conducted to determine this (lines 164-167). Do all of these genes act at phase I in the fissioning behavior like *hox3s*?

It is unclear from the description of the model how the authors suggest that the radial parenchymal and epidermal cells interface with the nervous system to control fissioning behavior.

It is interesting that *hox3a* has both positive and negative roles in the overall process. Could it be that *hox3(RNAi)* animals "arrest" midway in the fission behavior stage, in other words that unsuccessful attempts at fission lead to the additional compression planes in these animals? Also, it is interesting

that *hox3b* is expressed in the prepharyngeal region (Currie 2016). In the *hox3* phenotype of excess segments is there an enrichment of excess segments preferentially in the anterior?

It would be very useful to include plots from digiworm, or the equivalent, showing how specifically expressed are some of the target genes. Also, based the currently available resources, it is currently more straightforward as a reader to find such information if *ddv6* names are presented.

The diagram in Fig S8g suggests animals do not have fission planes prior to the RNAi experiment, but according to the authors prior work, I believe they do exist even in small animals. If so, I would suggest modifying the diagram accordingly.

Please indicate primer sequences and gene contig IDs used in the study.

Figure 8f needs a negative control condition as shown.

Some readers could be confused by the term radial here, given that similar cell populations have previously been termed lateral.

The terms in the figures used to describe the lateral/radial cell populations are likely to be difficult to interpret to readers from outside the field. For example, in Fig4g the terms 8/13, 6/10, 10 and 5 might at first appear to scorings or some other numbers. Calling them cluster8/13, cluster 6/10, etc in the figure would help convey the message more effectively.

Malinowski et al 2017 also recently described a mechanical model underlying fissioning in *D. Japonica* and observed pulsations reminiscent to those described here, and I think this study's contribution should be cited. In general, the manuscript only minimally cites important related work from planarians that is relevant for the overall problems of body segmentation, patterning, and scaling.

Reviewer #3 (Remarks to the Author):

Comments to the authors:

The manuscript entitled "Hox genes regulate asexual reproductive behavior and tissue segmentation in adult animals" by Alejandro Sánchez Alvarado and his colleagues shows achievement of adult asexual reproduction by Hox genes in a planarian, *Schmidtea mediterranea*. Asexual reproduction of *Schmidtea mediterranea* is initiated by "growth and pre-segmentation in the adult body" and accomplished by subsequent "fission behavior composed of contortion and stretch of the body". Silencing Hox3 genes resulted in supernumerary segments and also increased frequency of fission behavior, especially contortion. Furthermore, silencing of *post2b* caused elimination of both segmentation and fission behavior. Although opposing roles of Hox3 genes and *post2b* were thus observed in segmentation and fission behavior, silencing of those genes resulted in a decrease of fission frequency. The authors also showed a network of downstream effector genes involved in regulation of asexual reproduction by these Hox genes.

All results are clear, and satisfactorily addressed. The role of Hox genes in the postnatal body is very impressive and give a new insight into new function of Hox genes. I believe that this manuscript is worth publishing in Nature Communications.

Response to Reviewers:

We are thankful to the reviewers for their constructive criticism and overall positive feedback. We have carefully considered their comments and have taken a focused effort to address each one of the reviewer's concerns. In the process, we have resolved a clearer mechanism underlying *Hox* gene regulation of asexual reproduction that has resulted in several new text sections, an expanded Figure 4, and the addition of Extended Data Figures 9 and 10. We thank the reviewers for their thoughtful suggestions, which we believe have substantially improved the scope and quality of the work.

Reviewer #1 (Remarks to the Author):

This is an interesting follow-up paper of a previous one entitled "The role of Wnt and TGF β signaling in the control of size-dependent behavior of the flatworm *Schmidtea mediterranea*" (Nature 572, 655–659). In that first paper, the authors have described a process of asexual reproduction in planarians during which adult animals stretch and contract their tail tissue and produce posterior tissue fragments that regenerate into intact animals. This process is depending on the size of the animals, it involves active biomechanical forces and it seems to be under the control of a nervous subsystem acting downstream of Wnt/TGF β signaling. In the paper under review the authors analyzed now the role of *hox* genes in the same process. They identified by gene silencing 5 *hox* genes that are required for the fission process, the flatworms form of asexual reproduction. Silencing of *hox3* and *post2b* revealed an opposing role of both *hox* genes: Silencing of *hox3* resulted in supernumerary segments, while silencing of *post2b* inhibited fission completely. Downstream effector genes that mediate *hox* gene regulation were also identified and they revealed a detailed picture of postembryonic function of *hox* genes in planarians. The manuscript is well written and provides a substantial data set, which allows the statement that *hox* genes are active in the fission process of asexually reproducing planarians. There are two principal comments on the manuscript.

(i) It is certainly interesting to learn more about the function of *hox* genes in adult organisms, although it is not surprising that *hox* genes are active in adult planarians. Those are asexually reproducing animals, in which embryonic features are always preserved in adult animals, as they are necessary for the patterning of new structures. My suggestion is to tone down this argument.

(ii) But more importantly, the authors have previously explained fission through the effect of Wnt and TGF signaling on the nervous system and patterning processes. While it's obvious that transcription factors act on a different cellular control level than signaling factors (i.e., Wnt/TGF β signaling), the interesting and important question is now how are the two mechanisms linked. The authors address this point at the end of the manuscript and come to the conclusion that "that a subset of Wnt-independent Hox genes mediate specific functions in asexual reproduction and work in concert with Wnt-dependent patterning and regeneration programs". But what is the proof that these five hox genes and especially hox3 and post2b, are Wnt independent? These important conclusions are not yet sufficiently substantiated. I am also wondering why the authors put much effort in the identification of the target genes of the identified hox genes instead of trying to uncover the interrelationship of Wnt/TGF β signaling and hox genes in this segmentation-like process in planarians. It would be very helpful to have seen some ATACseq data or promotor studies for the five hox genes identified in order to bridge the gap between both papers. Nevertheless, I support publication of this interesting manuscript.

Reviewer #1 (Remarks to the Author):

Point #1-1: (i) It is certainly interesting to learn more about the function of hox genes in adult organisms, although it is not surprising that hox genes are active in adult planarians. Those are asexually reproducing animals, in which embryonic features are always preserved in adult animals, as they are necessary for the patterning of new structures. My suggestion is to tone down this argument.

Point #1-1 Response: We believe this is a reasonable suggestion for the paper and may explain why *Hox* mediated regulation of segmentation is an adult stage developmental program in this organism and an embryonic program in others. We have toned down the emphasis on adult tissue functions and inserted the following text to acknowledge the reviewer's point:

Lines 55-58

“As asexually reproducing animals, planarians preserve many embryonic developmental programs within their adult tissues, including the ability to establish and maintain the polarization and patterning of the body plan along the A/P axis^{12,13}.”

Point #1-2: (ii) But more importantly, the authors have previously explained fission through the effect of Wnt and TGF signaling on the nervous system and patterning processes. While it's obvious that transcription factors act on a different cellular control level than signaling factors (i.e., Wnt/TGFβ signaling), the interesting and important question is now how are the two mechanisms linked. The authors address this point at the end of the manuscript and come to the conclusion that “that a subset of Wnt-independent Hox genes mediate specific functions in asexual reproduction and work in concert with Wnt-dependent patterning and regeneration programs”. But what is the proof that these five hox genes and especially *hox3* and *post2b*, are Wnt independent? These important conclusions are not yet sufficiently substantiated. I am also wondering why the authors put much effort in the identification of the target genes of the identified hox genes instead of trying to uncover the interrelationship of Wnt/TGFβ signaling and hox genes in this segmentation-like process in planarians. It would be very helpful to have seen some ATACseq data or promotor studies for the five hox genes identified in order to bridge the gap between both papers.

Point #1-2 Response: We believe the reviewer makes several valid points here. First, upon reconsideration, our claim of Wnt-independence of *hox3* and *post2b* is too premature as it is based on negative data from two published RNAseq experiments (Tewari et. al 2019, Reuter et. al. 2015). We have therefore altered the discussion section to better reflect this evidence (see text revision below). Secondly, we see that we did not do a proper job in explaining our rationale for undertaking the *de novo* screen of *Hox* effectors to uncover the downstream mechanisms of action rather than investigating the connection to Wnt/TGFbeta. We have therefore inserted two new paragraphs that presents the published evidence that indicated to us that the *post2b* and *hox3* were operating by an independent mechanism.

Lines 160-190

“Our findings indicate that *Hox* genes regulate asexual reproduction via the emergence and modulation of A/P segmentation and behavior in adult animals. We previously identified requirements for the Wnt signaling components β -*catenin*, *apc*, *dshB*, and *wnt11-6* as well as the TGF- β signaling components *actR-1* and *smad2/3* in the regulation of fission behavior⁴. Their functions coincided with their roles in the size-dependent patterning of putative mechanosensory neurons expressing *gabrg3L-2* and *pkd1L-2*, genes with roles in the inhibition of fission activity^{4,26}. The interplay between *Hox* and *Wnt* signaling is an ancient feature of axial patterning broadly conserved throughout phyla^{22,27,28}. Additionally, TGF- β and/or BMP signaling has been shown to regulate *Hox* gene expression and collaborate with HOX proteins for target gene regulation^{29,30}. This raises the question as to the extent to which *hox1*, *hox3a*, *hox3b*, *lox5b*, and/or *post2b* interact with Wnt/TGF- β signaling in the regulation of asexual reproduction.

We explored the potential interplay between Wnt/TGF- β and the identified fission regulatory *Hox* genes in planarian asexual reproduction. While knockdown of β -*catenin* alters expression of multiple *Hox* genes, including *hox4b* (aka *hoxD*), *lox5a*, *post2c*, and *post2d* (aka *abdBa*), there is no significant effect on the expression of *hox1*, *lox5b*, *hox3a*, *hox3b*, or *post2b*^{22,31}. Additionally, while the expression domain of *post2b* along the lateral edge of the animal resembles that of *gabrg3L-2* and *pkd1L-2*, single-cell RNAseq (scRNAseq) analysis indicates that they are expressed within non-overlapping cell clusters^{18,26,32} (**ED Figure 7e-g**). Finally, the RNAi phenotypes of Wnt/TGF- β pathway components are distinct from that of newly identified fission-regulatory *Hox* genes. Animals treated with *hox1*, *hox3a*, *hox3b*, *lox5b*, or *post2b* RNAi lack detectable anterior-posterior regeneration phenotypes (**Fig1e,g**) and knockdown of *hox3a*, *hox3b*, or *post2b* alters fission segmentation (**Fig2**). In contrast, knockdown of fission regulatory Wnt/TGF- β components yields anterior-posterior regeneration defects and has little to no effect on fission segmentation^{4,31,33-36}. The lack of an obvious functional integration of fission-regulatory *Hox* genes into the Wnt/TGF- β pathway motivated a *de novo* investigation of the mechanism by which *Hox* genes regulate asexual reproduction.”

Additionally, we have further resolved the mechanism of action of *post2b* (See Figure 4). Finally, in our discussion section, we emphasize that future studies examining the

interconnectedness of the Wnt/TGFbeta mechanism we previously reported and the novel *Hox* mechanism we report herein are warranted.

Lines 331-342

“In conclusion, our study identifies roles for planarian *Hox* genes as mediators of adult tissue segmentation and fission behavior in addition to providing the first evidence that *Hox* genes function in asexual reproduction. In combination with prior work, our current findings indicate complementary roles for *Hox* and *Wnt/TGF-β* genes during planarian asexual reproduction⁴. The coordinate roles of *hox1*, *lox5b*, *hox3a*, *hox3b*, and *post2b* regulate segmentation for progeny tissue allocation, adhesion for posterior anchoring to substrates, and the execution of transverse fission behavior to ultimately separate tissue fragments that will regenerate into clonal progeny (**Fig4k**). In parallel, *Wnt/TGF-β* activity coordinates growth with size-dependent patterning of fission-modulatory mechanosensory neurons⁴. Given the preponderance of evidence for the interplay of *Hox* genes with the Wnt and TGF-β pathways^{22,27-30}, it would be of interest to further investigate the extent of their interdependence in the regulation of asexual reproduction.”

Reviewer #2 (Remarks to the Author):

This report identifies unexpected functions for Hox genes and their regulatory targets in the process of planarian asexual reproduction by fissioning. In this process, planaria rip themselves apart at specified locations on the AP axis, followed by regeneration to form new progeny. The authors had previously established assays for monitoring animal behavior before and during fission attempts, that unfissioned animals have latent “segments” of compressional mechanical sensitivity marking the sites of future fissioning, and that Wnt signaling and CNS activity are required for controlling the fissioning mechanism.

Here, Arnold et al examine Hox genes for potential roles in this process, finding three categories of defects following Hox gene RNAi. Inhibition of *hox1* and *lox5b* caused overall fission failure with no defects to compression planes (“segments”) and instead failure of the behavior of fissioning attempts. Inhibition of *hox3a/b* caused overall fission failure along with excess compression planes and increased frequency of attempts. Finally, inhibition of *post2b*, a laterally

expressed hox gene, disrupted compression planes entirely and also eliminated fission behavior. To understand the cellular mechanisms underlying these effects, the authors used RNAseq to identify transcripts dependent on the hox1, lox5b, hox3a/b and post2b genes. They inhibited 423 of ~1700 such identified transcripts and found 24 of these were required for fissioning, representing putative downstream targets of each of the initial Hox genes uncovered by the study. Hox-binding motifs could be identified in the regions around most of these targets, suggestive that they could be direct targets. Focusing on post2b targets, the authors then cross-referenced with cell atlas data to identify the cell types co-expressing post2d and the targets as representing two different putatively radial/lateral parenchymal cell types and a nearby epidermal cell type located at the D/V boundary. None of these factors was important for specifying compression plane segments but were instead important for fissioning behavior. Together the paper makes several important contributions, showing unexpected roles for Hox genes in controlling fission behavior and the planarian segments. The segmentation phenotype of post2b RNAi is particularly striking and reveals a new process in the segmentation mechanism. In addition, the analysis to identify targets involved a substantial RNAi screen, revealing novel regulators of the fissioning process. Altogether, the work makes a substantial step forward in understanding the fissioning mechanism which is a critical component of understanding growth and regeneration in these animals.

Comments:

The evidence at hand does not yet definitively show the involvement of the lateral cell populations in mediating fissioning behavior, given the small number of cells recovered by scRNAseq and lack of readily available information in the manuscript for the expression of these target genes. Showing in situ hybridizations verifying that lateral/radial expression of factors is dependent on post2b would offer significantly greater support for the model in 4g. Similarly, double-FISH experiments would verify the cell-type specific expression detected by drop-seq. As it stands, it is unclear from the data presented how specifically expressed are the targets and whether post2b regulation affects this expression.

The paper builds toward uncovering functions for the targets downstream of post2d mediating segmentation and fissioning behavior, but surprisingly none is required for generating compression planes (segments). What is the author's interpretation of these results? Is segmentation itself likely controlled by alternate post2d expressing cells, factors not included in the screen, or through redundant processes among these factors or by some other process? Based on the evidence from the screens so far, it would seem that fissioning behavior is modified by a larger number of inputs than physical segmentation. Is it possible

that the design of the screen scoring could have prevented identification of factors acting downstream of *hox3* or *post2d* specifically in the segmentation process?

The data showing lack of evidence for an effect of inhibiting *post2d* downstream genes on segmentation needs to be shown because of the central role this plays in the model. It is also unclear as stated, exactly what experiment was conducted to determine this (lines 164-167). Do all of these genes act at phase I in the fissioning behavior like *hox3s*?

It is unclear from the description of the model how the authors suggest that the radial parenchymal and epidermal cells interface with the nervous system to control fissioning behavior.

It is interesting that *hox3a* has both positive and negative roles in the overall process. Could it be that *hox3*(RNAi) animals “arrest” midway in the fission behavior stage, in other words that unsuccessful attempts at fission lead to the additional compression planes in these animals? Also, it is interesting that *hox3b* is expressed in the prepharyngeal region (Currie 2016). In the *hox3* phenotype of excess segments is there an enrichment of excess segments preferentially in the anterior?

It would be very useful to include plots from *digiworm*, or the equivalent, showing how specifically expressed are some of the target genes. Also, based the currently available resources, it is currently more straightforward as a reader to find such information if *ddv6* names are presented.

The diagram in Fig S8g suggests animals do not have fission planes prior to the RNAi experiment, but according to the authors prior work, I believe they do exist even in small animals. If so, I would suggest modifying the diagram accordingly.

Please indicate primer sequences and gene contig IDs used in the study.

Figure 8f needs a negative control condition as shown.

Some readers could be confused by the term radial here, given that similar cell populations have previously been termed lateral.

The terms in the figures used to describe the lateral/radial cell populations are likely to be difficult to interpret to readers from outside the field. For example, in Fig4g the terms 8/13, 6/10, 10 and 5 might at first appear to scorings or some

other numbers. Calling them cluster8/13, cluster 6/10, etc in the figure would help convey the message more effectively.

Malinowski et al 2017 also recently described a mechanical model underlying fissioning in *D. Japonica* and observed pulsations reminiscent to those described here, and I think this study's contribution should be cited. In general, the manuscript only minimally cites important related work from planarians that is relevant for the overall problems of body segmentation, patterning, and scaling.

Reviewer #2 (Remarks to the Author):

Point#2-1: The evidence at hand does not yet definitively show the involvement of the lateral cell populations in mediating fissioning behavior, given the small number of cells recovered by scRNAseq and lack of readily available information in the manuscript for the expression of these target genes. Showing *in situ* hybridizations verifying that lateral/radial expression of factors is dependent on *post2b* would offer significantly greater support for the model in 4g.

Point#2-1 Response: We agree that WISH analysis would offer greater support for our model. We have therefore performed WISH for the *post2b* effector genes. We obtained sufficient signal to resolve the expression of *rcn-1*, *syt1*, *ifb*, and *laminA/C* (ED Figure 9F). Additionally, we have performed WISH for these markers in control, *post2b*, and *hox3* RNAi treated animals, revealing that expression of these genes within the lateral zone is *post2b*-dependent (Figure 4f). This is consistent with previous data supporting *post2b*-dependent expression of *ifb* in lateral epidermal cells (Wurtzel et. al. 2017). The following text has been added.

Lines 247-262

“WISH analysis delineated an inner layer of *rcn-1*⁺ *syt1*⁺ cells and an outer layer of *laminA/C*⁺ and *ifb*⁺ cells along the lateral edge of the animals that paralleled published *post2b* expression patterns (**Fig4e**, **ED Fig9f**). These scRNAseq and *in situ* analyses indicate that *post2b* and its respective downstream effector genes are co-expressed within specific planarian cell populations.

We next set out to determine whether *post2b* regulated the expression of these cognate downstream effectors within the identified cell populations. We used WISH to analyze gene expression in *post2b* RNAi-treated worms. Knockdown of *post2b*, but not *hox3*, eliminated lateral edge staining of *rcn-1*, *syt1*, *lmnAC*, and *ifb* (**Fig4f**). These results are consistent with previous reports that *post2b* regulates *ifb* expression in planarian lateral epidermal cells²¹. Notably, RNAi targeting of *post2b* did not affect *rcn-1* expression in parapharyngeal cells outside the domain of *post2b* expression¹⁸. Collectively, this expression analysis indicates that *post2b* maintains the expression of genes within lateral parenchymal and epidermal cell populations that are required for asexual reproduction.”

Point#2-2: Similarly, double-FISH experiments would verify the cell-type specific expression detected by drop-seq.

Point#2-2 Response: We have performed double WISH for *rcn-1* with *syt1*, *ifb*, and *laminA/C* (Figure 4e). Additionally, we cite previously published double in situ WISH that supports the co-expression of *rcn-1* with *mag-1* within subepidermal gland cells of the marginal adhesive gland (Zayas et. al. 2010). Furthermore, we cite previous work in *Macrostomum* demonstrating intermediate filament expression specifically within the epidermal anchor cells of the adhesive gland (Lengerer et. al 2014). We have added the following text:

Lines 285-287

“Cross-referencing scRNASeq data, *in situ* hybridization data, and published literature revealed that the lateral parenchymal and epidermal cell populations are *mag-1*⁺ viscid gland cells and *ifb*⁺ epithelial anchor cells of the marginal adhesive gland (MAG)^{21,40}.”

Point#2-3: As it stands, it is unclear from the data presented how specifically expressed are the targets and whether *post2b* regulation affects this expression.

Point#2-3 Response: We hope the reviewer will agree that the inclusion of the WISH data from Fig 4e, f, ED Fig 9f and additional references clarifies the cell-type specific expression and *post2b*-dependence of the effectors *rcn-1*, *syt1*, *ifb*, and *ImnAC*.

Point#2-4: The paper builds toward uncovering functions for the targets downstream of *post2d* mediating segmentation and fissioning behavior, but surprisingly none is required for generating compression planes (segments). What is the author's interpretation of these results? Is segmentation itself likely controlled by alternate *post2d* expressing cells, factors not included in the screen, or through redundant processes among these factors or by some other process? Based on the evidence from the screens so far, it would seem that fissioning behavior is modified by a larger number of inputs than physical segmentation. Is it possible that the design of the screen scoring could have prevented identification of factors acting downstream of *hox3* or *post2d* specifically in the segmentation process?

Point#2-4 Response: We believe the reviewer is referring to *post2b* and not *post2d* in this and some other comments. Currently, our best explanation for the lack of uncovered downstream segmentation effectors is as follows. The *post2b* RNAi samples from the RNAseq experiment are from 8 dsRNA feedings. While this is sufficient to manifest the behavioral phenotype, it is not long enough of a treatment to manifest the loss of compression planes. This phenotype appears after 15-18 bacterial RNAi feedings. We only learned of the additional *post2b* segmentation phenotype once we had performed these longer-term experiments and after the RNAseq experiment was already completed. As a result, it is likely that the DEGs from *post2b* RNAi in this experiment do not include genes required for segmentation and thus was not part of our initial screen. We have included the following text to clarify this point:

Lines 108-113

“Notably, the loss of segmentation following *post2b* RNAi occurred in animals fed greater than 15 times with bacteria expressing *post2b* dsRNA but not in animals fed only nine times with purified dsRNA (**data not shown**). Based on the feeding duration, this observation suggests that the directly or indirectly affected transcripts, proteins,

and/or cell populations required for fission segmentation have a turnover time greater than one month.”

With respect to *hox3*, there is a cluster of genes upregulated specifically in *hox3a* and *hox3b* RNAi that correlates with the increase in segments. Genes from this cluster were not present in the initial library for RNAi screening. Many have since been cloned and will be part of a future study of the mechanism of *hox3*-dependent segmentation.

Point#2-5: The data showing lack of evidence for an effect of inhibiting *post2d* downstream genes on segmentation needs to be shown because of the central role this plays in the model. It is also unclear as stated, exactly what experiment was conducted to determine this (lines 164-167).

Point#2-5 Response: We apologize because this is poorly worded in the manuscript and leads to some ambiguity. To be clear, the experiments that we performed to test whether the effectors regulate segmentation are RNAi followed by compression. The “data not shown” experiment refers to compression we did for animals in the primary RNAi screen. Given the large number of RNAi conditions in this screen, phenotypes were noted for the lack or appearance of fission segments rather than each being imaged. This screen included all of the identified *post2b* effectors and segmentation was observed in all cases. We therefore referred to these results as “data not shown”. In a follow up experiment, we performed RNAi on a subset of *post2b* RNAi effectors and then compressed and imaged them. Figure 4g shows the compression of RNAi animals for *post2b* effectors *u-02422*, *plg*, *syt-1*, *ifb*, *lmnAC*, *post2a*, and *rcn-1*. We have removed the “data not shown” reference to eliminate the ambiguity. See text below.

Lines 265-267

“Surprisingly, RNAi of tested *post2b* effectors did not phenocopy the loss of segmentation observed in *post2b* RNAi animals (**Fig4g**).”

Point#2-6: Do all of these genes act at phase I in the fissioning behavior like *hox3s*?

Point#2-6 Response: To resolve the effects of *post2b* effectors on fissioning behavior, we performed timelapse imaging and analysis on RNAi-treated animals (Figure 4h, ED Fig 10). Individual knockdown significantly reduced both fission behavior initiation and success for nearly all tested effectors (Fig 4h ED Fig10h). Effects on fission behavior duration were mixed with RNAi knockdowns leading to slight increases, slight decreases, or no change on fission behavior duration (ED Fig 10i). Altogether, these data indicate a different phenotype than *hox3* RNAi which results in more fission initiations and a much longer fission event duration due to a phase I stall (Figure 3). The following text has been added:

Lines 268-274

“Knockdown of *rcn-1*, *plg1*, *syt1*, *ifb*, *lmnAC*, or *post2a* significantly reduced or eliminated fission attempts, largely phenocopying *post2b* RNAi (**Fig4h, ED Figure 10, Fig3b-e, Video S11-S17**). RNAi phenotypes of most effectors were not as strong as that of *post2b*, suggesting that multiple effectors work in concert to give rise to the *post2b* phenotype. These findings indicate that *post2b*-mediated regulation of effector gene expression within these lateral parenchymal and epidermal cell populations is required for fission behavior.”

Point#2-7: It is unclear from the description of the model how the authors suggest that the radial parenchymal and epidermal cells interface with the nervous system to control fissioning behavior.

Point#2-7 Response: We agree with the reviewer that this explanation for the role of the effectors in fission behavior was not entirely satisfying to us as well. With further research and experimentation, we have arrived at a much more plausible and better supported model. The radial parenchymal and epidermal cells are the adhesive glands and anchor cells of the marginal adhesive organ. This duo gland system is required for the animal to adhere to substrates. Since the first step of a fission event is adherence of the posterior to the substrate, the animals never initiate a fission behavior because they cannot perform this critical first step. By analyzing electron micrographs, we confirm that

RNAi of *post2b* compromises the structure and mucosal secretions of the marginal adhesive gland. We have expanded the text in our revision to explain the new findings and model as follow:

Lines 276-329

“Our screening analysis identified multiple *Hox* gene effectors required for asexual reproduction (**Fig4**). ScRNA analysis and WISH hybridization indicate that the effectors required for fission are expressed within multiple cell populations in the adult animal³² (**Fig4, ED Fig9**). These observations suggest that gene function within multiple tissues and cell types, in addition to the CNS, are required for asexual reproduction and even different aspects of fission behavior⁴. But, in the case of the identified *post2b* effectors, how are gene functions within lateral parenchymal and epidermal cells required for fission behavior initiation?

Cross-referencing scRNASeq data, *in situ* hybridization data, and published literature revealed that the lateral parenchymal and epidermal cell populations are *mag-1*⁺ viscid gland cells and *ifb*⁺ epithelial anchor cells of the marginal adhesive gland (MAG)^{21,40}. Within this duo gland organ conserved across platyhelminthes the coordinate functions of adhesive gland cells (viscid gland cells), releasing cells, and anchor cells mediate transient adhesion to substrates^{41,42}. Viscid gland cells within the parenchyme expel their adhesive secretions onto the animal’s surface via long cellular processes that cross the basal lamina and travel through epithelial anchor cells to outer pores surrounded by adhesive papilla^{41,42}. In *Macrostomum lignano*, RNAi targeting of the anchor cell-specific intermediate filament gene *macif1* results in severe morphological alterations in anchor cells and an inability to adhere to substrates⁴². Assuming conservation of gene functions, RNAi targeting of the anchor cell-specific planarian intermediate filament gene *ifb* likely compromises the structure and function of the epithelial secretory pore. Furthermore, the *post2b* effector *syt1* is expressed in viscid gland cells and is homologous to *synaptotagmin*, a calcium sensor for regulated exocytosis⁴³. Additionally, the *post2b* effect *rcn-1*, also expressed in viscid gland cells, is homologous to *reticulocalbin*, a calcium-binding protein of the secretory pathway⁴⁴. Based on the functions of these homologous genes, we predict that RNAi targeting of *rcn-1* and *syt1* likely compromises calcium regulated exocytosis of adhesive secretions

from the viscid gland. Notably, RNAi targeting *post2b* or downstream effectors expressed within the marginal adhesive organ (*rcn-1*, *plg1*, *syt1*, *ifb*, *lmnAC*, and *post2a*) results in animals constantly drifting across substrates while adopting a resting position (**Videos S7, S12-17**). This phenotype is consistent with a defect in substrate adherence and inability to stop cilia-driven locomotion and remain stationary. Given that fission behavior begins with the anchoring of the worm's posterior end to a substrate and requires functional effectors specifically expressed within the MAG, we conclude that control of MAG constitutes a key facet of planarian asexual reproduction.

To investigate the function of *post2b* in the marginal adhesive organ, we used electron microscopy to characterize the cellular structure of the viscid gland and adhesive cells in *post2b* RNAi animals. Scanning electron micrographs (SEM) of an amputated planarian tail region revealed a distinct mucus stripe along the lateral edge epithelial surface, corresponding to the adhesive secretions of the MAG (**Fig4i, ED Fig11a**). Knockdown of *post2b* eliminated or vastly reduced this stripe of mucosal secretions. Notably, the exposed lateral edge epithelial surface of *post2b* RNAi animals had expected complements of cilia and microvilli but lacked detectable adhesive papillae. Analysis of transverse sections in control animals resolved parenchymal viscid gland cells extending processes that crossed the basal lamina to connect to adhesive papillae-covered pores on the epithelial anchor cell surface (**Fig4j, ED Fig11b**). Consistent with our exterior SEM analysis, *post2b* RNAi animals lacked a distinct anchor cell region in the lateral margin epithelium and displayed a reduced number of viscid gland cell extensions crossing the basal lamina and terminating at the epithelial surface. Of the few viscid gland cells detected, all lacked adhesive papillae around the pore termini. Altogether, our findings indicate that *post2b* functions in the MAG are required for initiation of planarian fission behavior.”

Point#2-8: It is interesting that *hox3a* has both positive and negative roles in the overall process. Could it be that *hox3*(RNAi) animals “arrest” midway in the fission behavior stage, in other words that unsuccessful attempts at fission lead to the additional compression planes in these animals?

Point#2-8 Response: This is an interesting idea and would suggest that behavior and segmentation are dependent processes. Although, there are some issues with this

hypothesis. *APC* RNAi increases fission attempts without increasing fission segments (Arnold and Benham-Pyle et. al. 2019). Also, *post2b* effector RNAi eliminates fission behavior without a decrease in fission segments. This would argue that segmentation and behavior are independent, but the issue is not yet settled.

Point#2-9: Also, it is interesting that *hox3b* is expressed in the prepharyngeal region (Currie 2016). In the *hox3* phenotype of excess segments is there an enrichment of excess segments preferentially in the anterior?

Point#2-9 Response: We have quantitated the difference in the number of fission segments in the pre-pharyngeal versus the post-pharyngeal regions of the animal. We do observe that *hox3* RNAi animals have increased fission planes in the anterior (New Fig2c). We have added the following text:

Lines 115-117

“The increased fission segmentation following *hox3* RNAi was slightly biased in the region anterior of the pharynx, coinciding with the expression domain of *hox3b*¹⁸ (Fig 2c).”

Point#2-10: It would be very useful to include plots from digiworm, or the equivalent, showing how specifically expressed are some of the target genes.

Point#2-10 Response: We agree with the reviewer and have included t-SNE plots from digiworm for the key *post2b* effectors in the study (ED Fig9a-d)

Point#2-11: Also, based the currently available resources, it is currently more straightforward as a reader to find such information if *ddv6* names are presented.

We agree with the reviewer and have included the *ddv6* of all the genes in Table 1 and referenced these *dd* IDs in the text and Figures in place of SMEDID.

Point#2-12: The diagram in Fig S8g suggests animals do not have fission planes prior to the RNAi experiment, but according to the authors prior work, I believe they do exist even in small animals. If so, I would suggest modifying the diagram accordingly.

Point#2-12 Response: To clarify, in planaria 1-2mm long there are no detectable fission planes (Arnold and Benham-Pyle et. al. 2019; Fig3b,c). The cartoon of the small animal in Fig1a and new Fig4k represents this size. In the old figure FigS8g, the cartoon depicts the *post2b* RNAi experiment in which RNAi treated animals lack fission planes in all sizes.

Point#2-13: Please indicate primer sequences and gene contig IDs used in the study.

Point#2-13 Response: We have created Table 1 which lists the SMEDID, dd_Smed_v6, Genbank Accession Number, primer sequences, and Literature References for all of the *Hox* genes and identified effector genes in this study.

Point#2-14: Figure 8f needs a negative control condition as shown.

Point#2-14 Response: We have repeated the experiment and included a negative control condition in addition to increasing the n of each RNAi condition (New Fig4g).

Point#2-15: Some readers could be confused by the term radial here, given that similar cell populations have previously been termed lateral.

Point#2-15 Response: We agree and have changed this term to lateral when referenced in the text.

Point#2-16: The terms in the figures used to describe the lateral/radial cell populations are likely to be difficult to interpret to readers from outside the field. For example, in Fig4g the terms 8/13, 6/10, 10 and 5 might at first appear to scorings or some other numbers. Calling them cluster8/13, cluster 6/10, etc in the figure would help covey the

message more effectively.

Point#2-16 Response: Given our new *in situ* data and identification of the cells as part of the marginal adhesive organ, the cartoon in Old Fig4g is no longer necessary and we have removed it.

Point#2-17: Malinowski et al 2017 also recently described a mechanical model underlying fissioning in *D. Japonica* and observed pulsations reminiscent to those described here, and I think this study's contribution should be cited. In general, the manuscript only minimally cites important related work from planarians that is relevant for the overall problems of body segmentation, patterning, and scaling.

Point#2-17 Response: We thank the reviewer for this feedback. The initial submission of the manuscript required a short format with a maximum number of citations. We now have the opportunity to include additional background, context, and literature references. We have added several new sections to the introduction and increase literature citations throughout the manuscript. Our total number of references has increased from 31 to 57. With respect to Malinowski et al. 2017, we fully agree with the reviewer and have cited this study in reference to the pulsation phenotype. Below are some of the sections that have been added/revised in reference to important planarian work and other relevant literature:

Lines 29-69

“The generation of a complex multicellular organism from a fertilized oocyte has long fascinated developmental biologists¹. Remarkably, some animals can achieve this same feat independent of gametic fusion by generating an entire organism directly from their adult tissues. This plasticization of somatic tissues for asexual reproduction is a vital part of the life cycle of a diverse array of metazoans, including members of Porifera, Cnidaria, Acoela, Lophotrochozoa, and Deuterostomia². Furthermore, asexually reproductive organisms exhibit an unparalleled capacity for injury-induced regeneration, indicating that asexual reproduction constitutes a selective force for the persistence of regenerative capacity³. Therefore, resolution of molecular mechanisms mediating

asexual reproduction has the potential to inform our understanding of regulated tissue plasticity, clonal animal propagation, and the evolution of animal regeneration.

We recently uncovered and characterized the size-dependent behaviors and adult tissue structures underlying asexual reproduction in the planarian, *Schmidtea mediterranea*⁴ (**Fig1a**). These highly regenerative flatworms exist in both sexual and asexual biotypes. The latter reproduces solely through transverse fission – an asexual reproductive behavior in which torn posterior tissue fragments regenerate and give rise to clonal progeny⁵. Fission behavior is size-dependent, and its establishment and regulation coincide with growth-dependent patterning of the planarian central nervous system (CNS)⁴. Our study also revealed a cryptic form of segmentation that allocates tissue for fission progeny in coordination with the dynamic growth and de-growth of the adult planarian body plan^{4,6}. Presently, the molecular mechanisms mediating this segmentation and its coupling to asexual reproductive behavior are still unknown.

The ancestral roles of *Hox* genes, a family of transcription factors with evolutionarily conserved functions in embryonic head to tail patterning, are tightly linked to the emergence of segmented animal body plans^{7,8}. Additionally, *Hox* gene functions in neuronal development throughout taxa suggest potentially ancient roles in the evolution of the nervous system⁹⁻¹¹. As asexually reproducing animals, planarians preserve many embryonic developmental programs within their adult tissues, including the ability to establish and maintain the polarization and patterning of the body plan along the A/P axis^{12,13}. We therefore hypothesized that *Hox* genes play a role in A/P directed tissue segmentation and/or transverse fission behavior underlying asexual reproduction in adult planarians.

There has been a great deal of interest in understanding *Hox* gene function in planaria. Planarian *Hox* genes were first cloned over 30 years ago^{14,15}. Since then, multiple studies have characterized the spatio-temporal regulation of planarian *Hox* genes during homeostasis and regeneration^{14,16-18}. With the introduction of RNA-mediated genetic interference (RNAi)¹⁹, many investigators made extensive efforts to perturb planarian *Hox* gene function (both individually and combinatorially) with no discernible phenotypic defects reported^{18,20,21}. Only recently has a requirement for the planarian *Hox* gene *Post2d* in proper tail regeneration been reported²². Thus, to date, the functions of *Hox* genes in planaria remain largely a mystery.”

Lines 83-88

“Notably, *hox3b* is enriched in a band of expression anterior to the pharynx in 4-5mm animals¹⁸. This region overlaps with the location of newly arisen fission planes as planarians grow from 3 to 5mm in length⁴. Additionally, *hox1*, *hox3a*, *hox4b*, *post2d*, and *post2b* exhibit overlapping expression with the cholinergic neuron marker, *chat* in the planarian brain and nerve cord¹⁸.”

Lines 155-157

“This peristalsis was previously observed in live imaging of fissioning *S. mediterranea*, and a similar process was observed in fissioning *Dugesia japonica*^{4,5}.”

Lines 344-351

“Our study also elucidates *Hox* gene-mediated segmentation within historically designated unsegmented flatworms⁴⁵. In combination with the discovery of *Hox*-mediated segmentation in the basal metazoan *Nematostella vectensis*, these findings support ancient roles for *Hox* genes in the regulation of body segmentation in addition to their well-established roles in segmental patterning⁴⁶. Finally, we identify roles for *Hox* genes in multiple aspects of asexual reproductive behavior, building upon the known roles for *Hox* genes in developing nervous systems and functions linked with specific animal behaviors^{9-11,47}”

Reviewer #3 (Remarks to the Author):

Comments to the authors:

The manuscript entitled “Hox genes regulate asexual reproductive behavior and tissue segmentation in adult animals” by Alejandro Sánchez Alvarado and his colleagues shows achievement of adult asexual reproduction by Hox genes in a planarian, *Schmidtea mediterranea*. Asexual reproduction of *Schmidtea mediterranea* is initiated by “growth and pre-segmentation in the adult body” and accomplished by subsequent “fission behavior composed of contortion and stretch of the body”. Silencing Hox3 genes resulted in supernumerary segments

and also increased frequency of fission behavior, especially contortion. Furthermore, silencing of post2b caused elimination of both segmentation and fission behavior. Although opposing roles of Hox3 genes and post2b were thus observed in segmentation and fission behavior, silencing of those genes resulted in a decrease of fission frequency. The authors also showed a network of downstream effector genes involved in regulation of asexual reproduction by these Hox genes. All results are clear, and satisfactorily addressed. The role of Hox genes in the postnatal body is very impressive and give a new insight into new function of Hox genes. I believe that this manuscript is worth publishing in Nature Communications.

Reviewer #3 (Remarks to the Author):

All results are clear, and satisfactorily addressed. The role of Hox genes in the postnatal body is very impressive and give a new insight into new function of Hox genes. I believe that this manuscript is worth publishing in Nature Communications.

We appreciate the reviewer's positive evaluation and hope that the reviewer agrees that the revisions and new additions further improve the quality of the manuscript.

** See Nature Research's author and refere

REVIEWERS' COMMENTS

Reviewer #1 (Remarks to the Author):

The authors carefully revised their manuscript and all points I raised were clarified and in some cases new data were added, e.g., expanded Fig. 4 and, in response to reviewer 2, expanded Figs. 9 and 10. In their work, the authors convincingly demonstrated how hox genes form new body parts during asexual reproduction of planarians by fission, a process in which planarians separate body fragments from themselves along the anterior-posterior axis, which then regenerate to form new, complete worms.

The significance of this new and excellent work by Arnold et al now lies in the fact that in classical zoology, flatworms have always been considered a group of non-segmented animals. But this apparently does not apply at the molecular level, because Arnold et al show a segmentation-like process (i.e. fission) that is mediated by hox genes. Interestingly, sea anemones (polyps of Cnidaria) can also reproduce asexually by fission, involving also Hox genes. This suggests an ancient role for Hox genes in regulating body segmentation that is older than previously thought, and it provides an idea of how animals that possessed segmental patterning along the AP axis may have arisen in the evolution of Bilateria: Bilaterian body segments might have evolved along the AP axis by hox gene activity and downstream of Wnt signaling. This is reminiscent to what was shown in the work of Arnold et al for planarians with their impressive "fission" behavior, but without a separation of body fragments (= segments).

Reviewer #2 (Remarks to the Author):

This revision now gives rigorous and compelling support that the lateral edge adhesive/anchor cell system controlled by the post2b Hox gene enables the adhesive steps needed for fissioning behavior. The addition of new functional assays, cell type expression of Hox gene targets and EM characterization of the RNAi conditions added considerable mechanistic depth to the conclusions. All of my comments are resolved. The paper defines a novel use for Hox genes, and I congratulate the authors on this outstanding contribution to the field.